# STABLE VIDEO INFINITY: INFINITE-LENGTH VIDEO GENERATION WITH ERROR RECYCLING

**Wuyang Li    Wentao Pan    Po-Chien Luan    Yang Gao    Alexandre Alahi**
**VITA@EPFL**
**Project Page:** *https://stable-video-infinity.github.io/homepage/*

## ABSTRACT

We propose **Stable Video Infinity (SVI)** that can generate non-looping, ultra-long videos with stable visual quality, while supporting per-clip prompt control and multi-modal conditioning. While existing long-video methods attempt to *mitigate accumulated errors* via handcrafted anti-drifting (e.g., modified noise scheduler, frame anchoring), they remain limited to single-prompt extrapolation, producing homogeneous scenes with repetitive motions. We identify that the fundamental challenge extends beyond error accumulation to a critical discrepancy between the training assumption (seeing clean data) and the test-time autoregressive reality (conditioning on self-generated, error-prone outputs). To bridge this hypothesis gap, SVI incorporates **Error-Recycling Fine-Tuning**, a new type of efficient training that recycles the Diffusion Transformer (DiT)'s self-generated errors into supervisory prompts, thereby encouraging DiT to *actively identify and correct its own errors*. This is achieved by injecting, collecting, and banking errors through closed-loop recycling, autoregressively learning from error-injected feedback. Specifically, we (i) inject historical errors made by DiT to intervene on clean inputs, simulating error-accumulated trajectories in flow matching; (ii) efficiently approximate predictions with one-step bidirectional integration and calculate errors with residuals; (iii) dynamically bank errors into replay memory across discretized timesteps, which are resampled for new input. SVI is able to scale videos from seconds to infinite durations with no additional inference cost, while remaining compatible with diverse conditions (e.g., audio, skeleton, and text streams). We evaluate SVI on three benchmarks, including consistent, creative, and conditional settings, thoroughly verifying its versatility and state-of-the-art role.

## 1 INTRODUCTION

> *Failure is simply the opportunity to begin again, this time more intelligently.*
> — HENRY FORD

With the scaling of models and data, the video Diffusion Transformer (DiT)  (Wang et al., 2025a; Kong et al., 2024; Liu et al., 2024; Hong et al., 2023) has made great strides in synthesizing realistic, temporally coherent videos, supporting in-the-wild content creation. While achieving great success, this community suffers from a limited video length, typically 5 seconds (Wang et al., 2025a). This is mainly caused by the open challenge of **error accumulation**, a.k.a., drifting: after autoregressively conditioning on the previously generated, predictive errors will compound over time, leading to progressive degradation in image fidelity, motion stability, and semantic controllability (Fig. 1.a).

In this context, existing solutions can be divided into three trends, including **(i) noise modification**, augmenting and modifying the noise schedule to reduce the past-frame dependency (Chen et al., 2024; Ruhe et al., 2024), **(ii) frame anchoring** using the error-free reference image (Henschel et al., 2025) as anchors to reduce the dependency of error-included ones, and **(iii) improved sampling** like masked-noise guidance (Song et al., 2025) and anti-drifting sampling (Zhang & Agrawala, 2025).

However, existing methods primarily aim to alleviate rather than correct accumulated errors, which leads to two key limitations: **constrained length** (generally 10 seconds up to about 1 minute) and the **scene homogeneity bias** with repetitive motion. Practically, most methods essentially extrapolate the original clips controlled by a single prompt, rather than creating truly long-form videos

Figure 1: Comparison among (a) video generative DiT, (b) restoration DiT, and (c) our Stable Video Infinity regarding the scheme (*row 1*), training-test hypothesis gap (*row 2*), and outcome (*row 3*).

that the prompt stream storylines can easily control. Consequently, current solutions do not satisfy many creative real-world demands, such as short-form filming that requires plausible, frequent scene changes or creation of hour-scale online presentations.

To tackle this, we aim to treat the cause rather than the symptom, seeking to fundamentally correct accumulated error itself rather than merely alleviating its effects. By observing the artifacts caused by errors (see Fig. 6), we empirically find that they closely align with common degradation types, such as blur and color shift within the image restoration community. Given the state-of-the-art role of DiT in low-level vision, these degradations should not be difficult for much larger video DiT (e.g., 14B) with more substantial capacity. Surprisingly, the opposite holds in practice: Why are these powerful models highly susceptible to such errors, leading to a severe and rapid collapse?

We uncover that the fundamental challenge lies in the *hypothesis gap between the training and test*. In training generative DiT (Fig. 1a), historical trajectories of flow matching are assumed to be error-free. However, this is easily broken in test since the model autoregressively uses previous generations with predictive errors, which is mathematically clarified in Sec. 3. In contrast, for restoration DiT (Fig. 1b), both training and test assume error-injected inputs, ensuring error robustness. Hence, to bridge this gap, we start a new perspective: *recycling self-generated errors as supervisory prompts, encouraging DiT to autoregressively correct its own mistakes via error feedback*.

In this work, we propose **Stable Video Infinity (SVI)** to generate ultra-long videos with stable quality and non-looping motions. In Fig. 1c, SVI employs a novel **Error-Recycling Fine-Tuning** that repurposes the DiT's self-generated errors as supervisory signals, thereby enabling the model to iteratively refine its outputs through autoregressive error feedback. Specifically, we (i) intervene on clean input by injecting historical errors to simulate degradation, (ii) approximate the predictions and calculate errors via one-step integration bidirectionally, and (iii) dynamically save and selectively resample errors across discretized timesteps from replay memory. By doing so, we can efficiently unleash the restoration ability in video DiT, actively correcting errors in generation. Additionally, SVI has several emergent advantages over previous works. ***Data light***: only small-scale data required for LoRA fine-tuning; ***Efficient***: zero additional inference cost; ***Versatile***: supporting in-the-wild control signals, e.g., audio and skeleton (Fig. 7c).

In summary, our contributions are fivefold. (1) SVI breaks the length limit of videos from seconds to infinity by actively correcting errors. (2) We systematically analyze the training-test hypothesis gap in long video generation and theoretically formulate two types of errors. (3) To bridge this gap, an error-recycling fine-tuning is proposed to dynamically calculate, save, and selectively inject errors to clean inputs, predicting error-recycled velocity. (4) We extend SVI into a family of models for different applications, e.g., talking and dancing (see Fig. 7c). (5) We propose comprehensive benchmarks with short/long consistent/creative settings, aligning with diverse real user needs.

## 2 RELATED WORK

**Video Generation in the Wild.** With the scaling of model (Hu et al., 2025) and data (Weissenborn et al., 2020), commercial-grade video generative models (Liu et al., 2024; Brooks et al., 2024; Yang et al., 2024c; Blattmann et al., 2023; Ho et al., 2022; Lin et al., 2024), such as Wan (Wang et al., 2025a) and Hunyuan (Chen et al., 2025), have made significant progress in producing high-quality short videos. Based on this, the community has pursued a flourishing line of secondary creation for diverse objectives, introducing task-oriented controls, e.g., audio, skeleton, to generate desired

content like talking (Kong et al., 2025; Chen et al., 2025), dancing (Wang et al., 2025b), navigation (Agarwal et al., 2025; Hassan et al., 2025), gaming (Yu et al., 2025; Che et al., 2025). Despite promising progress, the short length remains an open challenge, limiting its practical applications.

**Long Video Generation.** Generating long videos is an open problem due to error accumulation, prompting three trends of solutions. (i) Modified scheduler: aims to improve the ODE solver with error robustness. Some works (Qiu et al., 2024; Høeg et al., 2024) extend videos via noise rescheduling. Some other works (Chen et al., 2024; Ruhe et al., 2024; Song et al., 2025) modify and augment the noise schedule to reduce dependence on past frames. (ii) Frame anchoring: uses the clean image as a consistent reference, including tailored anchor designs (Henschel et al., 2025; Weng et al., 2024) and planning-based optimization (Brooks et al., 2024; Zhao et al., 2024; Yang et al., 2024b). (iii) Error-robust architecture: improve long-range consistency autoregressively with bidirectional distillation (Yin et al., 2025; Huang et al., 2025), distributed generation (Tan et al., 2024), and anti-drifting sampling (Zhang & Agrawala, 2025; Gu et al., 2025), improved attention Kodaira et al. (2025); Lu et al. (2024), mixture of context (Cai et al., 2025). In this work, we start a new perspective by recycling the errors made by DiT itself, encouraging DiT to actively correct errors cyclically.

## 3 PRELIMINARIES AND MOTIVATION

### 3.1 ERROR-FREE HYPOTHESIS IN LONG VIDEO TRAINING

**Notation.** We use *hat* $(\hat{\cdot})$, *tilde* $(\tilde{\cdot})$, and the *superscript-free* symbol $(\cdot)$ to represent *model predicted*, *error-injected*, and *clean (error-free)* variables, respectively, in following sections.

**Training.** Flow matching enables continuous-time generation for DiT. To delve into the essential challenge of the training-test hypothesis gap, we start from error-free flow matching (Fig. 2a) for image-to-video training, which aims to learn the model solving ODE from the joint noise and reference image distribution $X_{\text{noi}}^{\text{img}}$ to video latent $X_{\text{vid}}$. In training, assuming *error-free video latent* $X_{\text{vid}}$, noise $X_{\text{noi}} \sim \mathcal{N}(0, I)$, timestep $t \in [0, 1]$, *error-free reference image* $X_{\text{img}}$, and optional multimodal condition $C$, the training objective is denoted as:

$$\mathcal{L} = \mathbb{E}_{X_{\text{noi}}, X_{\text{vid}}, X_{\text{img}}, C, t} \big| u(X_t, X_{\text{img}}, C, t; \theta) - V_t \big|^2, \quad (1)$$

where $X_t = t \cdot X_{\text{vid}} + (1-t) \cdot X_{\text{noi}}$ is the intermediate state, $\hat{V}_t = u(X_t, X_{\text{img}}, C, t; \theta)$ is the predicted velocity with model $\theta$, $V_t = \frac{dX_t}{dt} = X_{\text{vid}} - X_{\text{noi}}$ is the ground-truth.

**Test.** The generated video latent is obtained at $t = 1$ with $X_1 = X_{\text{vid}}$ using $N_{\text{test}}$ sampling from noise $X_0$. In practice, this process can be achieved by discretizing the unit interval $0 = t_0 < t_1 < \cdots < t_{N_{\text{test}}} = 1$ and applying a numerical ODE solver (Esser et al., 2024) for the step-wise integration:

$$X_{t_{k+1}} = X_{t_k} + (t_{k+1} - t_k) \cdot u(X_{t_k}, X_{\text{img}}, t_k; \theta), \quad (2)$$

where $k < N_{\text{test}} - 1$, and $X_{t_{k+1}}$ is next-step generation.

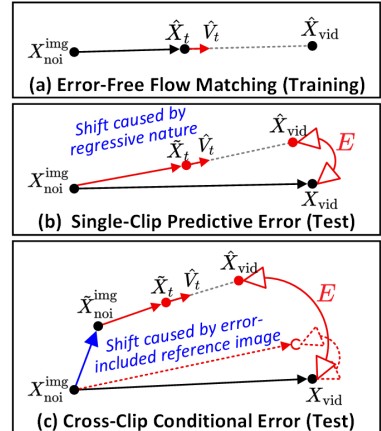

(a) Error-Free Flow Matching (Training)

(b) Single-Clip Predictive Error (Test)

(c) Cross-Clip Conditional Error (Test)

Figure 2: **Training-test hypotheses gap.** (a) Training assumes historical trajectories and an intermediate stage free of errors, which are easily broken in test by two errors. (b) Predictive error caused by the regressive nature affects the trajectory end $X_{\text{vid}}$. (c) Conditional error caused by error-included images also affects start $\tilde{X}_{\text{noi}}^{\text{img}}$.

### 3.2 ERROR-CORRUPTED INFERENCE IN LONG VIDEO GENERATION

In the test, we uncover that two types of errors will appear due to the error-free training hypothesis.

**Single-Clip Predictive Error**. In Eq. 1, the training assumes $X_t$ obtained via a clean latent $X_{\text{vid}}$ with correct historical trajectory. However, in inference (Fig. 2b), this hypothesis is easily broken, since $\tilde{X}_t$ is obtained from a predictive trajectory with inherent errors. Due to the Mean-Square-Error (MSE) regressive nature, this will lead to an eternal existent difference $E_t$ between the predicted velocity $\hat{V}_t = u(\tilde{X}_t, X_{\text{img}}, C, t; \theta)$ and ground-truth $V_t = u(X_t, X_{\text{img}}, C, t; \theta)$. This shift is gradually accumulated at each sampling step, which reuses the integrated velocity from previous steps. Consequently, for the independent generation, the small step-wise errors integrate over the ODE trajectory

in Eq. 2, producing a shifted predicted latent $\hat{X}_{\text{vid}}$, which is defined as *single-clip sampling error*: $E = \hat{X}_{\text{vid}} - X_{\text{vid}}$. Conceptually, $E$ is typically small with adversarial attack nature (Goodfellow et al., 2015) and causes negligible perceptual degradation in short video generation.

**Cross-Clip Conditional Error.** In Fig. 2c, when generating subsequent clips autoregressively, the model uses **error-included** frame $\tilde{X}_{\text{img}}$ from $\hat{X}_{\text{vid}}$ (Fig. 2b) instead of the *clean one* $X_{\text{img}}$ used in training (Eq. 1), leading to a shift in the trajectory start from $X_{\text{vid}}^{\text{img}}$ to $\tilde{X}_{\text{vid}}^{\text{img}}$. As Eq. 1 is optimized with *clean input*, these *error-corrupted samples* $\tilde{X}_{\text{vid}}^{\text{img}}$ are out-of-distribution regarding clean training data, which will severely confuse DiT in predicting $\hat{V}_t = u(\tilde{X}_t, \tilde{X}_{\text{img}}, C, t; \theta)$. Due to its large gap with desired velocity $V_t = u(X_t, X_{\text{img}}, C, t; \theta)$ in training, this error will cause a more biased prediction $X_{\text{vid}}$, where we define this accumulated error as *cross-clip conditional error*: $E = \hat{X}_{\text{vid}} - X_{\text{vid}}$. Here, $\hat{X}_{\text{vid}}$ is solved by integrating $\hat{V}_t$ with error-corrupted $\tilde{X}_{\text{img}}$.

**Error Accumulation and Amplification.** In autoregressive cross-clip conditioning, these two types of error are *accumulated and reinforce each other*: predictive error induces drift in the generated video latent, which magnifies the error at the trajectory start and, in turn, further increases predictive error. This feedback loop can rapidly cause catastrophic degradation of generated videos.

### 3.3 Bridging the Training-Test Hypothesis Gap

In summary, the essential challenge for long video generation is the hypothesis gap between the error-free training and error-included inference. More harmfully, DiTs tend to *accumulate and amplify these errors* in the autoregressive condition, rather than correcting them. To bridge this gap, we aim to break the error-free training hypothesis with **Error-Recycling Fine-Tuning (ERFT)** to stabilize the DiT in long generation, which recycles the DiT's self-generated errors into supervisory prompts, thereby encouraging the model to correct its own mistakes from autoregressive error feedback. Mathematically, the error-recycling objective is defined as follows ,

> Given error-injected and clean inputs at random timestep $t$, **ERFT** aims to predict an **error-recycled velocity**: $V_t^{\text{rcy}} = u(\tilde{X}_t, \tilde{X}_{\text{img}}, C, t; \theta)$ and $V_t^{\text{rcy}} = u(X_t, X_{\text{img}}, C, t; \theta)$, respectively, to stabilize DiT in autoregressive generation, which consistently points to clean latents $X_{\text{vid}}$, regardless the correctness of the current state $\tilde{X}_t$ and historical trajectory before $t$.

We achieve this via a closed-loop fine-tuning: inject errors $E^1$ made by DiT to simulate degradation (Sec. 4.1), calculate and save errors $E$ (Sec. 4.2), and dynamically bank and resamples errors $E$ for new input (Sec. 4.3), finally optimizing error-recycled velocity $V_t^{\text{rcy}}$ (Sec. 4.4).

## 4 Stable Video Infinity

### 4.1 Error-Recycling Fine-tuning

Given a clean video clip $\{I_i\}_{i=1}^{T_{\text{vid}}}$ and reference image $I_i$, we extract the video $X_{\text{vid}}$ and image $X_{\text{img}}$ (typically using padding) latent via 3D VAE that both $\in \mathbb{R}^{C \times T \times H \times W}$. Then, we randomly sample a noise $X_{\text{noi}} \in \mathbb{R}^{C \times T \times H \times W}$ drawn from $\mathcal{N}(0, I)$ and a timestep $t \in \mathcal{T}_{\text{tra}}$ to train the video DiT.

**Error Injection.** Unlike existing works assuming clean input, we aim to simulate error-accumulated degradation occurring in inference. Given clean input $X_{\text{vid}}, X_{\text{noi}}, X_{\text{img}}$, we design three types of errors accordingly: $E_{\text{vid}}, E_{\text{noi}}, E_{\text{img}}$. These errors are resampled from the memory banks $\mathcal{B}_{\text{vid}}, \mathcal{B}_{\text{noi}}$, which are explained in the next sections. Then, we inject errors into clean inputs probabilistically:

$$\tilde{X}_{\text{vid}} = X_{\text{vid}} + \mathbb{I}_{\text{vid}} \cdot E_{\text{vid}}, \quad \tilde{X}_{\text{noi}} = X_{\text{noi}} + \mathbb{I}_{\text{noi}} \cdot E_{\text{noi}}, \quad \tilde{X}_{\text{img}} = X_{\text{img}} + \mathbb{I}_{\text{img}} \cdot E_{\text{img}}. \quad (3)$$

Here, $\mathbb{I}_* = 1$, w.p. $p_*$ else 0 controls probability $p_*$ of error injection. This design aims to simulate the randomness and complexity of error accumulation appearing in any inference timestep. To preserve the generation ability with corrected errors, we set a probability $p = 0.5$ using the error-free input. Hence, the ultimate input $\tilde{X}_t$ sent to DiT blocks is denoted as $\tilde{X}_t = \text{Concat}(\tilde{X}_t, \tilde{X}_{\text{img}})$,

---

[1]Considering duality, we use noise $E_{\text{noi}}$ and latent error $E_{\text{vid}}$ bidirectionally for theoretical completeness.

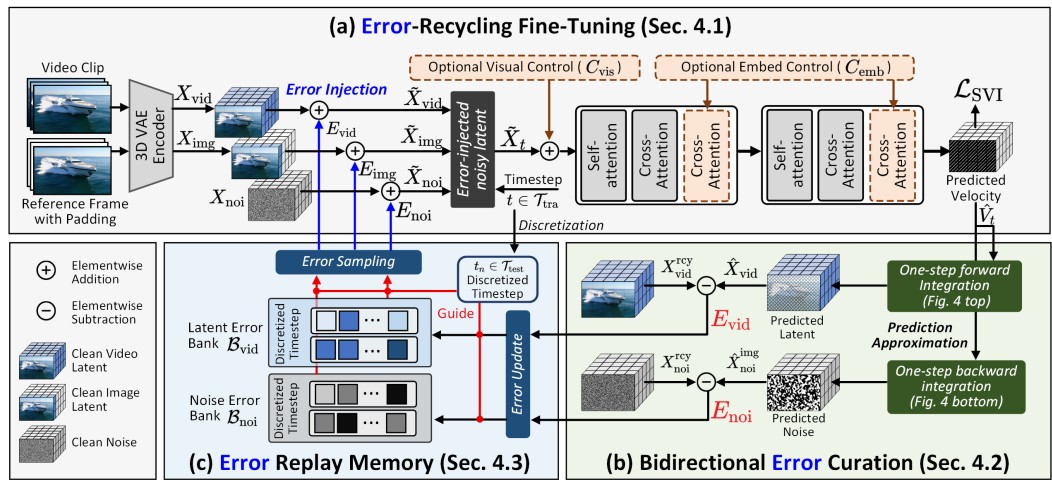

Figure 3: **Stable Video Infinity**. We (a) inject errors into clean latent to break the error-free hypothesis, (b) approximate predictions via one-step integration to calculate bidirectional errors, and (c) dynamically bank and resample errors from memory for clean inputs, in a closed-loop cycling.

where $\tilde{X}_t = t\tilde{X}_{\text{vid}} + (1-t)\tilde{X}_{\text{noi}}$ is noisy video latent with errors. This error-injection can fundamentally break the previous error-free hypothesis in Eq. 1, serving to bridge the train-test gap.

**Control Injection and Velocity Prediction.** To satisfy in-the-wild applications, we propose to extend SVI with extra controls $C = \{C_{\text{vis}}, C_{\text{emb}}\}$, justified in Fig. 7c. *(a) $C_{\text{vis}}$ is the visual condition* ensuring the spatial-level control on videos, e.g., the skeleton, which are injected at the tokenized input via element-wise addition. $C_{\text{vis}}$ can achieve precise control of spatial composition, serving tasks like dance animation. *(b) $C_{\text{emb}}$ is the embedding condition* for the multi-modal control without spatial constraints, e.g., text and audio for talking animation. $C_{\text{emb}}$ is injected via specific cross-attention layers in DiT blocks. Hence, the error-injected input $\tilde{X}_t$ is tokenized, optionally with $C_{\text{vis}}$, and sent to DiT blocks with optional $C_{\text{emb}}$ to predict the velocity: $\hat{V}_t = u(\tilde{X}_t, \tilde{X}_{\text{img}}, C, t; \theta)$.

## 4.2 BIDIRECTIONAL ERROR CURATION

Given the velocity $\hat{V}_t$, we approximate error-embedded predictions by a single-step integration bidirectionally for efficient error curation, which avoids the prohibitive cost of solving full ODEs.

**Prediction Approximation.** To tackle the complexity of accumulated errors, we delve into different error-inject scenarios in Fig. 4 to calculate errors. Aligning with our main objective (Sec. 3.3), we define the ground-truth ***error-recycled velocity*** $V_t^{\text{rcy}}$ (green single arrow) *pointing to the error-free latent* $X_{\text{vid}}$, independent of the historical trajectory and current state. Then, with error-injected noisy latent $\tilde{X}_t$ and predicted velocity $\hat{V}_t$ (red single arrow), we can approximate video latent $\hat{X}_{\text{vid}}$ and conditioned noise $\hat{X}_{\text{noi}}^{\text{img}}$ via one-step forward and backward integration, respectively (*red dotted line*): $X_{\text{vid}} = \tilde{X}_t + \int_t^1 V_s \, ds; \quad X_{\text{noi}}^{\text{img}} = \tilde{X}_t - \int_0^t V_s \, ds$. Similarly, we deploy the integration on $\hat{V}_t$ to obtain the error-recycled latent and noise: $X_{\text{vid}}^{\text{rcy}} = \tilde{X}_t + \int_t^1 V_s^{\text{rcy}} \, ds, X_{\text{noi}}^{\text{rcy}} = \tilde{X}_t - \int_0^t V_s^{\text{rcy}} \, ds$.

**Error Calculation.** With approximated predictions and error-recycled ground-truth, we examine each error-injected case in Fig. 4. We first present a unified formulation applicable to all cases:

$$E_{\text{vid}} = \hat{X}_{\text{vid}} - X_{\text{vid}}^{\text{rcy}}, \quad E_{\text{noi}} = \hat{X}_{\text{noi}}^{\text{img}} - X_{\text{noi}}^{\text{rcy}}, \quad E_{\text{img}} = \text{Unif}_T(E_{\text{vid}}), \quad (4)$$

where, $\text{Unif}(\cdot)$ is uniform sampling at the temporal axis $T$ in the latent space. We then prove this unified error curation aligning with Sec. 3.3 by detailing each real case as follows.

*(a) No Injected Error.* This can simulate the initial *single-clip predictive error* that the predicted velocity $\hat{V}_t$ is shifted anytime. Here, we define the latent and noise error with residuals: $E_{\text{vid}} = \hat{X}_{\text{vid}} - X_{\text{vid}}$ and $E_{\text{noi}} = \hat{X}_{\text{noi}}^{\text{img}} - X_{\text{noi}}^{\text{img}}$, where we also have $X_{\text{vid}}^{\text{rcy}} = X_{\text{vid}}$ and $X_{\text{noi}}^{\text{rcy}} = X_{\text{noi}}^{\text{img}}$.

*(b) Error-Injected Start Point.* This can simulate the *cross-clip conditional error* when the error causes shifts at the beginning from $X_{\text{noi}}^{\text{img}}$ to $\tilde{X}_{\text{noi}}^{\text{img}}$. Here, we can intervene on clean $X_{\text{img}}$ or $X_{\text{noi}}$

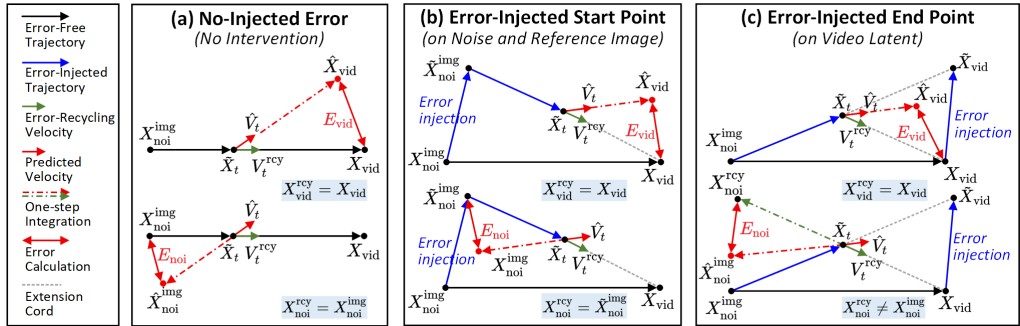

Figure 4: **Error calculation.** In different cases, the latent error $E_{\text{vid}}$ and noise error $E_{\text{noi}}$ are calculated by the one-step integration in the forward (top) and backward direction (bottom), respectively.

to simulate the error impacts. With error-recycled velocity, we can calculate bidirectional error with $E_{\text{vid}} = \hat{X}_{\text{vid}} - X_{\text{vid}}$ and $E_{\text{noi}} = \hat{X}_{\text{noi}}^{\text{img}} - \tilde{X}_{\text{noi}}^{\text{img}}$, where we have $X_{\text{vid}}^{\text{rcy}} = X_{\text{vid}}$ and $X_{\text{noi}}^{\text{rcy}} = \tilde{X}_{\text{noi}}^{\text{img}}$.

*(c) Error-Injected End Point.* This can simulate both error types from an accumulated perspective, where the previous integration points to a degraded generation $\tilde{X}_{\text{vid}}$. Our stabilized DiT is encouraged to verify and, if necessary, correct the wrong historical trajectory and wrong intermediate state $\tilde{X}_t$ towards degraded latent $\tilde{X}_{\text{vid}}$. To align with the error-recycled velocity, the errors can be written as $E_{\text{vid}} = \hat{X}_{\text{vid}} - X_{\text{vid}}$ and $E_{\text{noi}} = \hat{X}_{\text{noi}}^{\text{img}} - X_{\text{noi}}^{\text{rcy}}$, where we only have $X_{\text{vid}}^{\text{rcy}} = X_{\text{vid}}$.

### 4.3 ERROR REPLAY MEMORY

**Error Update.** We propose to dynamically save calculated errors $E_{\text{vid}}$ and $E_{\text{noi}}$ into two replay memory $\mathcal{B}_{\text{vid}}$ and $\mathcal{B}_{\text{noi}}$ according to timestep, respectively. To better reduce the train-test gap, we first discretize the training timestep $\mathcal{T}_{\text{tra}} = \{t_i\}_{i=1}^{N_{\text{tra}}}$ that is typically $N_{\text{tra}} = 1000$ by aligning it with the timestep used in test stage $\mathcal{T}_{\text{test}} = \{t_n\}_{n=1}^{N_{\text{test}}}$ with typically $N_{\text{test}} = 50$ as Eq. 2. Specifically, given a training timestep $t \in \mathcal{T}_{\text{tra}}$, we retrieve the nearest timestep grid $t_n$ in $\mathcal{T}_{\text{test}}$, and save each error $E_*$ into the corresponding location $B_{*,n}$ in bank $\mathcal{B}_* = \{B_{*,n}\}_{n=1}^{N_{\text{test}}}$, where $* = \{\text{vid}, \text{noi}\}$. Here, the timestep is a pointer pointing to the specific storage location. Considering the slow bank updates caused by the limited per-GPU sample, we design a warmup by saving errors with cross-machine gathering, inspired by Federated Learning (McMahan et al., 2017). To conserve the memory usage, we set an upper bound $Z = 500$ for the number of saved errors $|B_{*,n}| = Z$, justified in Appx C. When the specific bank $B_{i,*}$ is full, we replace the most similar one by measuring the $L_2$ distance between the new error $E_*$ and historical errors in $B_{*,n}$ to preserve error diversity.

**Error Sampling.** Considering specific roles of error terms, we design a selective sampling method based on the individual properties of each input term in flow-matching trajectories. Specifically, aligning with our error banking, we first discretize the training timestep $t$ sampled from $\mathcal{T}_{\text{tra}}$ to the test timestep $t_n \in \mathcal{T}_{\text{test}}$ by retrieving the nearest one. Then, for input terms $X_{\text{vid}}, X_{\text{noi}}, X_{\text{img}}$ in Eq. 3, the resampled errors are designed as follows accordingly,

$$E_{\text{vid}} = \text{Unif}(\mathcal{B}_{\text{vid},n}), \quad E_{\text{noi}} = \text{Unif}(\mathcal{B}_{\text{noi},n}), \quad E_{\text{img}} = \text{Unif}_T(\mathcal{B}_{\text{vid}}). \tag{5}$$

Here, $\text{Unif}(\mathcal{B}_{*,n})$ is uniform sampling conducted on the memory bank $B_{*,n} \in \mathcal{B}_*$ for the timestep $t_n$, and $\text{Unif}_T$ is performed across two dimensions: the whole timestep in the noise scheduler and the temporal axis of the video. The rationale for each selective strategy is explained as follows.

*(a) Video Latent Error* $E_{\text{vid}}$ is uniformly sampled from the timestep-aligned bank $\mathcal{B}_{\text{vid},n}$, because the step-wise errors predominantly depend on the current timestep $t_n$ throughout the trajectory. We also empirically find that degradation types are highly correlated with sampling steps.

*(b) Noise Error* $E_{\text{noi}}$, following the video latent, is sampled uniformly within the same timesteps from $B_{\text{noi},n}$, considering the duality between the noise (start) and latent (end).

*(c) Image Latent Error* $E_{\text{img}}$ is sampled from the video bank to align with the cross-clip autoregression, i.e., the generated frame serves as the reference image in the next-clip generation. Unlike the step-wise error, the reference image is obtained by integrating over all timesteps, which accumulates errors over the entire trajectory. To simulate this complexity, we sample $E_{\text{img}}$ across timesteps independently of the current $t_n$, because the error may occur and accumulate at any timestep.

| Models | Generated Scenes | Subject Consistency | Background Consistency | Aesthetic Quality | Imaging Quality | Dynamic Degree | Motion Smoothness |
|---|---|---|---|---|---|---|---|
| *Consistent Video Generation (single text prompt without scene transitions)* | | | | | | | |
| Wan 2.1 | Single | 87.03% | 92.45% | 56.40% | 65.70% | 12.68% | 98.51% |
| StreamingT2V | Single | 84.79% | 89.27% | 56.81% | 66.41% | **57.04%** | 99.00% |
| HistoryGuidance | Single | 83.77% | 90.90% | 40.42% | 55.48% | 4.93% | 99.38% |
| FramePack | Single | 93.08% | 94.72% | 63.57% | 66.72% | 7.75% | **99.57%** |
| SVI-Shot (Ours) | Single | **98.13%** | **98.19%** | **63.84%** | **71.88%** | 17.61% | 98.93% |
| *Ultra-Long Consistent Video Generation (single text prompt without scene transitions)* | | | | | | | |
| Wan 2.1 | Single | 80.00% | 87.27% | 56.19% | 65.37% | 14.29% | 98.74% |
| StreamingT2V | Single | 66.32% | 77.62% | 40.49% | 55.18% | **85.71%** | 95.60% |
| HistoryGuidance | Single | 64.84% | 80.51% | 29.84% | 50.41% | 7.14% | 99.42% |
| FramePack | Single | 79.37% | 86.64% | 55.66% | 57.61% | 0.00% | **99.63%** |
| SVI-Shot (Ours) | Single | **97.50%** | **97.89%** | **65.75%** | **71.54%** | 21.43% | 98.81% |
| *Creative Video Generation (text prompt stream with scene transitions)* | | | | | | | |
| Wan 2.1 | Multiple | 81.44% | 89.81% | 51.33% | 53.09% | 61.97% | 98.57% |
| SVI-Film (Ours) | Multiple | 84.25% | 90.85% | 55.25% | 59.97% | **62.68%** | 98.69% |
| StreamingT2V | Single | 81.01% | 88.47% | 52.20% | 58.05% | 61.97% | 98.96% |
| HistoryGuidance | Single | 84.12% | 91.21% | 38.40% | 52.31% | 7.75% | 99.39% |
| FramePack | Single | 85.62% | 91.22% | **59.41%** | 59.44% | 9.15% | **99.49%** |
| SVI-Shot (Ours) | Single | **93.52%** | **95.86%** | 58.07% | **62.81%** | 55.63% | 98.42% |
| *Ultra-Long Creative Video Generation (text prompt stream with scene transitions)* | | | | | | | |
| Wan 2.1 | Multiple | 67.85% | 83.45% | 46.68% | 43.36% | 57.14% | 98.56% |
| SVI-Film (Ours) | Multiple | 69.84% | 84.46% | 51.22% | 53.93% | **78.57%** | 98.50% |
| StreamingT2V | Single | 68.65% | 82.00% | 44.69% | 55.20% | **78.57%** | 96.95% |
| HistoryGuidance | Single | 62.58% | 81.97% | 28.66% | 47.68% | 7.14% | 99.36% |
| FramePack | Single | 70.95% | 83.46% | 52.39% | 53.72% | 0.00% | **99.48%** |
| SVI-Shot (Ours) | Single | **91.96%** | **95.04%** | **63.31%** | **65.25%** | 64.29% | 97.97% |

Table 1: Generic video generation with diverse settings. **Bold**, Underline highlights the highest, second highest, respectively. For more details on metrics, see (Huang et al., 2024b).

| Models | Sync-C ↑ | Sync-D ↓ | FVD ↓ |
|---|---|---|---|
| Wan 2.1 | 0.21 | 12.86 | 934 |
| MultiTalk | 1.26 | 9.57 | 520 |
| SVI-Talk (Ours) | **6.12** | **8.74** | **390** |

Table 2: Audio-conditioned long talk.

| Models | PSNR ↑ | SSIM ↑ | FVD ↓ |
|---|---|---|---|
| Wan 2.1 | 12.12 | 0.33 | 4099 |
| UniAnimate-DiT | 18.97 | 0.69 | 337 |
| SVI-Dance (Ours) | **20.01** | **0.71** | **299** |

Table 3: Skeleton-conditioned long dance.

## 4.4 OPTIMIZATION

To train Stable Video Infinity, we aim to predict error-recycled velocity $V_t^{\mathrm{rcy}} = X_{\mathrm{vid}} - \tilde{X}_{\mathrm{noi}}$ pointing to clean latent $X_{\mathrm{vid}}$ from error-injected inputs $\tilde{X}_{\mathrm{vid}}, \tilde{X}_{\mathrm{noi}}, \tilde{X}_{\mathrm{img}}$ obtained via Eq. 3. This aligns with our error-recycling objective (Sec. 3.3) in bridging the train-test hypothesis gap, denoted as follows,

$$\mathcal{L}_{\mathrm{SVI}} = \mathbb{E}_{\tilde{X}_{\mathrm{vid}}, \tilde{X}_{\mathrm{noi}}, \tilde{X}_{\mathrm{img}}, C, t} \big| u(\tilde{X}_t, \tilde{X}_{\mathrm{img}}, C, t; \theta) - V_t^{\mathrm{rcy}} \big|^2, \qquad (6)$$

where $\tilde{X}_t = t\tilde{X}_{\mathrm{vid}} + (1 - t)\tilde{X}_{\mathrm{noi}}$ is noisy latent with injected errors. To enable user flexibility, we only train LoRA. Our error-recycling tuning can actively correct the trajectory, unleashing DiT's restoration ability. Overall, in line with Henry Ford's quote at the beginning of this manuscript, we can rephrase it in the context of long video generation as: *"The accumulated error is simply an opportunity to begin again by recycling the errors. this time more stable — Stable Video Infinity ."*

## 5 EXPERIMENTS

**Benchmarks Setup.** We establish three benchmarks, consistent, creative, and conditional settings, for image and text-to-video generation (each has two variants), satisfying diverse industrial needs. *(a) Consistent Video Generation* aims to produce *50-sec* and *250-sec* (ultra-long) videos from an unchanging text prompt within one scene. *(b) Creative Video Generation* targets the needs of vloggers (e.g., TikTok) by emphasizing storytelling with plausible scene transitions. We develop an automatic engine via MLLM (see Appx. A) to generate prompt streams for videos of *50-sec* and *250-sec* (ultra-long) duration. *(c) Multimodal Conditional Generation* measures compatibility with extra conditions. We evaluate *300-sec* audio-guided talking and *50-sec* skeleton-guided dancing. **Metrics.** We use 6 core metrics from Vbench++ (Huang et al., 2024b) for global video quality. For specific conditional generation, we use Sync-C, Sync-D, FVD, PSNR, and SSIM metrics.

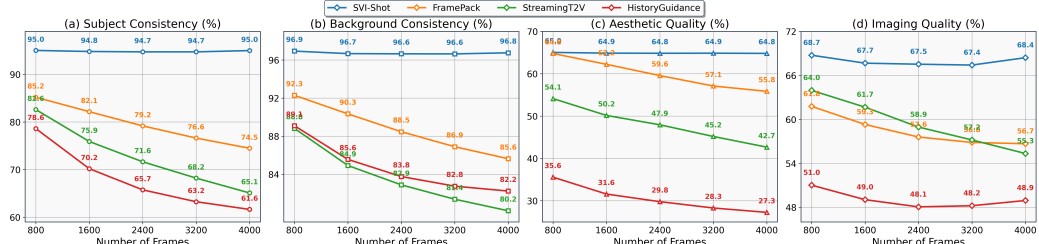

Figure 5: Stability comparison about video length. SVI is more stable without an obvious decrease.

| Method. | Sub. Cons. | Back. Cons. | Aest. Qual. | Img. Qual. |
|---------|-----------|-------------|-------------|------------|
| Wan 2.1 | 66.73% | 82.83% | 43.95% | 42.31% |
| SVI w/o $E_{\text{img}}$ | 73.82% | 84.21% | 49.58% | 57.63% |
| SVI w/o $E_{\text{noi}}$ | 94.22% | 94.87% | 59.80% | 69.90% |
| SVI w/o $E_{\text{vid}}$ | 93.56% | 95.01% | 58.99% | **71.50%** |
| SVI full | **94.69%** | **95.39%** | **61.88%** | 71.22% |

Table 4: Ablation study on each error term.

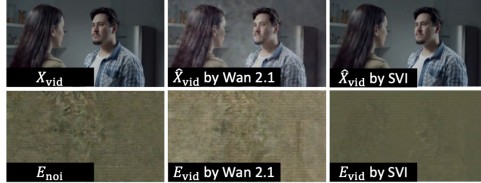

Figure 6: Error correction comparison.

**Implementations.** We deploy a family of SVI with minor modifications for diverse usages, which is only trained with 300-6k short videos (see Appx. D). *(a)* **SVI-Shot** aligns with the previous long video generation, focusing on a homogeneous scene. Here, in training, we conduct padding with a random image as the anchor to obtain the image latent, and replace it with the reference image in inference. *(b)* **SVI-Film** supports end-to-end long filming controlled with a storyline-based prompt stream. We use five motion frames and replace the padding frame with zero for the image latent. *(c)* **SVI-Talk** targets audio-conditioned human-talking videos, which extends SVI-Shot by injecting the audio-image cross-attention from (Kong et al., 2025). *(d)* **SVI-Dance** is for skeleton-guided dancing videos, which encodes the skeleton and injects it into input tokens following (Wang et al., 2025b). $E_{\text{img}}$, $E_{\text{vid}}$, and $E_{\text{noi}}$ are injected with probabilities 0.9, 0.9, and 0.01, respectively.

### 5.1 LONG VIDEO GENERATION IN THE WILD.

We compare with state-of-the-art methods: StreamingT2V (Henschel et al., 2025), HistoryGuidance (Song et al., 2025), FramePack (Zhang & Agrawala, 2025), and report Wan 2.1 (Wang et al., 2025a). We compare conditional generation with MultiTalk (Kong et al., 2025) and UniAnimate-DiT (Wang et al., 2025b) in audio-guided talking and skeleton-guided dancing, respectively.

**Consistent Video Generation.** In Tab. 1 top, SVI-Shot achieves the best results on most core metrics. Note that an abnormally large dynamic degree in this setting indicates uncontrollable motion degradation. Compared with FramePack, we give 5.05% and 3.37% gains on consistency, 5.16% gains on image quality. Most methods suffer from a large drop when extended to ultra-long videos, such as 7.03% and 13.71% subject consistency decrease for Wan 2.1 and FramePack. In contrast, SVI exhibits a negligible 0.63% decrease, while maintaining a satisfactory degree of dynamics.

**Creative Video Generation.** Tab. 1 bottom compares the long videos guided by a storyline-based prompt stream, which has frequent scene transitions. Note that existing long video works uniformly fail, as they cannot generate filming-level scene transitions with the prompt stream (see Fig. 7). SVI achieves the best consistency, quality, and a satisfactory dynamic degree with significant gains. This superiority is maintained when extended to ultra-long settings, showing the stability of SVI.

**Multimodal Conditional Generation.** In Tab. 2 and 3, we justify our adaptability with two typical conditions $C_{\text{emb}}$ with audio-guided talking, and $C_{\text{vis}}$ with skeleton-guided dancing mentioned in Sec. 4.1. It can be observed that SVI can effortlessly adapt to specific domains and enhance the state-of-the-art in long videos, verifying its versatility for in-the-wild generation.

### 5.2 FURTHER ANALYSIS

**Stability to Video Length.** Fig. 5 compares the robustness among the latest long video methods by measuring the consistency and quality. Unlike all existing works exhibiting a decreasing trend, SVI can maintain robust and high consistency and quality. This nature justifies the ability to generate an arbitrary length, which, for the first time, fundamentally corrects errors and breaks the time limit.

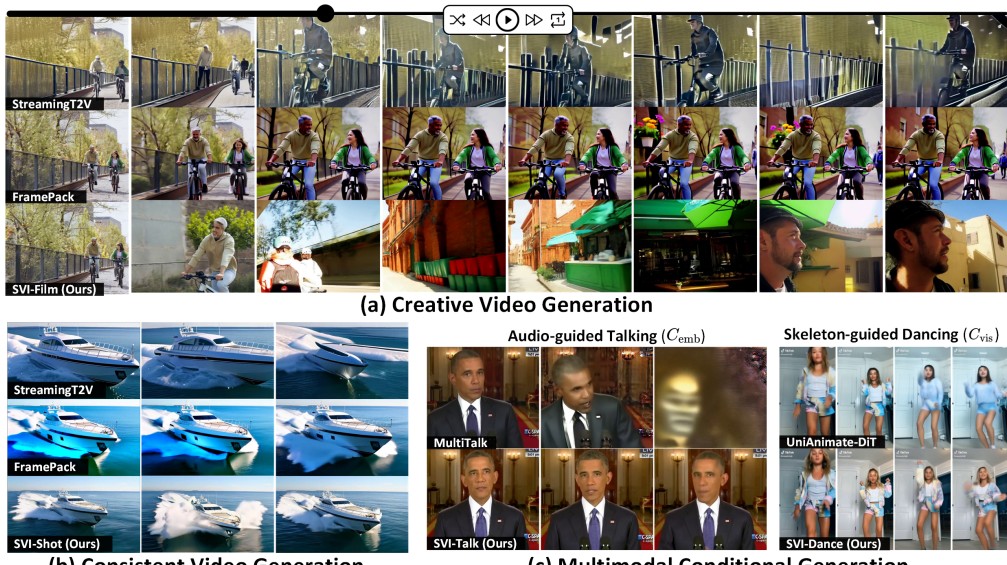

**(a) Creative Video Generation**

**(b) Consistent Video Generation**

**(c) Multimodal Conditional Generation**

Figure 7: Qualitative comparison with the best specific-domain methods (see videos in Appx E).

**Ablation Study.** In Tab. 4 top, we analyze each error with a 1-epoch tuning for proof-of-concept, revealing 2 critical messages. *(a)* Removing errors on the reference image $E_{\text{img}}$ gives a significant drop in all metrics. This indicates the primary role of intervening on the trajectory start (Fig. 4b) to simulate the error accumulation, justifying its direct and major role in solving the train-test hypothesis gap. *(b)* Injecting errors into the video latent $E_{\text{vid}}$ or noise $E_{\text{noi}}$ gives an auxiliary benefit in an indirect role compared with the major factor $E_{\text{img}}$. More ablations are in Appx. C.

**Error Visualization.** Fig. 6 visualizes the decoded error $E_{\text{vid}}$ and $E_{\text{noi}}$, and compares the prediction $\hat{E}_{\text{vid}}$, where we have two observations. *(a)* Video generators (Wan 2.1) are sensitive to the errors they make, leading to degraded prediction. This issue can be tackled by the error-recycling fine-tuning in SVI, achieving robustness of self-errors. *(b)* Injecting errors can well simulate the drifting ($X_{\text{vid}}$ by Wan 2.1), justifying the critical role of error recycling in bridging the train-test gap.

## 5.3 Qualitative Results

**Long Video Comparison.** Fig. 7a compares *Creative Video Generation* guided by a prompt stream. Existing works fail to achieve scene transitions with severe quality degradation. In contrast, SVI achieves smooth scene transitions, maintaining high visual fidelity and text-prompt following, which paves the way for end-to-end filming. In Fig. 7b, we compare *Consistent Video Generation* with an unchanged prompt, showing that existing methods suffer from color shifts, motion drifts, and degradation in static images. Differently, SVI generates temporally coherent videos with plausible consistency and dynamics. Fig. 7c compares *Multimodal Conditional Generation* between SVI and state-of-the-art counterparts in talking and dancing videos. Without being tailored to these domains, SVI can effortlessly tackle the drifting in long generation, justifying its effectiveness and transferability. Refer to the videos in Appx. E, including additional cross-domain adaptability.

**Extremely Ultira-Long Video Generation.** In Fig. 8, we explore the generation with 15-minute videos to verify the ultra-long, non-looping ability. ***Generic-Domain Generation:*** We use an LLM to generate 200 random prompts and perform end-to-end generation, conditioning each clip on its own prompt. This example shows SVI's potential to produce arbitrarily long videos across diverse, open-domain scenarios. ***Human-Centric Generation:*** We use a long audio track to drive a talking video generation. Each clip is conditioned on a different segment of the audio, so the motion does not loop. This example shows that SVI can maintain a stable identity within a single scene.

**Potential Long-Range Consistency.** We further fine-tune SVI-Shot on long-video datasets to enhance the long-range dependency and motion diversity. In Fig. 9, we observe an emergent long-range identity consistency happening in cross-clip conditioning without using identity-related text prompts. Here, the woman protagonist completely exits the current scene at 30 seconds, i.e., the end

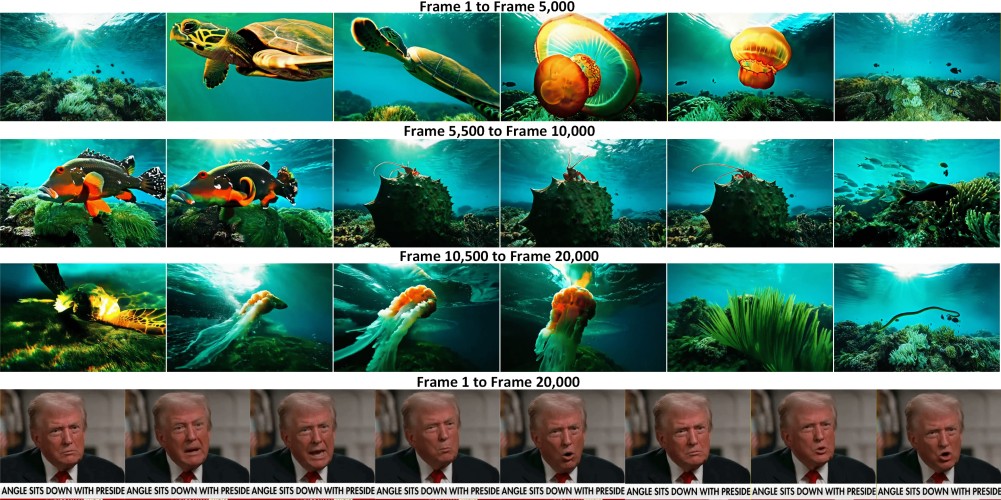

Figure 8: Extremely ultra-long generation results (10+ minutes). The videos are generated with long prompt streams (top) or different audio clips (bottom), showing the non-looping and diverse nature.

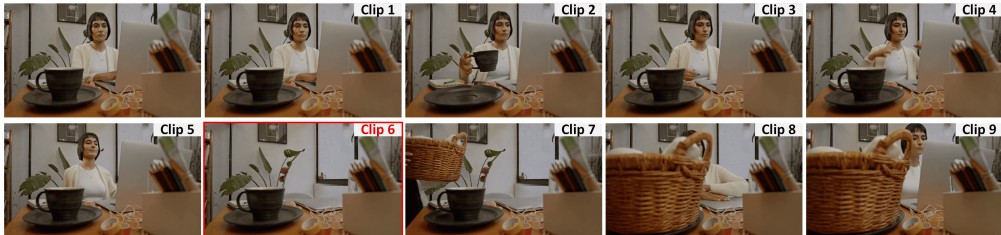

Figure 9: Visualization of last frames of generated video clips. The woman exits the current scene and then re-enters the next clip with similar personal information, e.g., haircut and clothes.

of the $6^{th}$ video clip, and then re-enters the next clip with similar personal information, e.g., haircut, and clothes, showing some potential for human-centric generation.

## 6 CONCLUSION AND DISCUSSION

We address the training–test hypothesis gap in long video generation, which leads to two forms of accumulated error (Sec . 3). To bridge this gap, we propose Stable Video Infinity to break the time limit by actively correcting the self-generated errors, which employs a novel Error-Recycling Fine-Tuning (Sec. 4) to autoregressively learn from error feedback. Unlike the error-free training assumption, SVI deliberately injects historical errors into clean inputs and learns to predict an error-recycled velocity that computes errors via bidirectional one-step integration, stores them in a replay memory, and selectively resamples them for new inputs. Across three benchmarks, SVI surpasses state-of-the-art methods on long, ultra-long, and conditional video generation.

**Discussing Train-Test Gap.** While sharing a similar goal, i.e., reducing the mismatch between training on clean contexts and inference under autoregressive, error-prone histories, the targets and strategies are different in some concurrent related works. Specifically, Self-Forcing (Huang et al., 2025) bridges the gap by performing autoregressive rollouts with KV cache during training and using distribution matching. Building on this, Rolling Forcing (Liu et al., 2025) addresses this gap by using a rolling-window joint denoising and an attention sink to suppress errors. LongLive (Yang et al., 2025) specializes in interactive, prompt-driven long videos by combining streaming long tuning with frame-sink. Some more recent works explore the RoPE and attention sink (Yesiltepe et al., 2025; Yi et al., 2025) for long generation in a training-free manner. In contrast to restructuring the video rollouts, SVI explicitly focuses on inference errors and encourages models to eliminate their own errors via error-recycling, achieving lightweight and flexible fine-tuning.

## ACKNOWLEDGEMENTS

This work was supported as part of the Swiss AI Initiative by a grant from the Swiss National Supercomputing Centre (CSCS) under project ID a144 on Alps. We would like to express our gratitude to Valentin Gerard, Tomasz Stanczyk, Megh Shukla, and Xiaoyuan Liu for insightful discussions.

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

## A    BENCHMARK SETUP

### A.1    AUTOMATIC PROMPT STREAM ENGINE

In this work, we study the creative generation using a storyline-driven text prompt stream. However, this is still an open problem in the community due to the lack of high-quality data and the laborious labeling process. To solve this issue and generate sufficient test data, we propose a *fully automated system for effortless, end-to-end short film production that streamlines creative video generation and evaluation*. The only user input required is the high-level subject specification (e.g., "dog" and "street"). The system then automatically retrieves and downloads relevant images and uses a Multimodal Large Language Model (MLLM) to generate a storyline-aligned text prompt stream for each video clip. These image–prompt pairs are subsequently fed into Stable Video Infinity to produce high-quality, narrative-driven short videos of unlimited length.

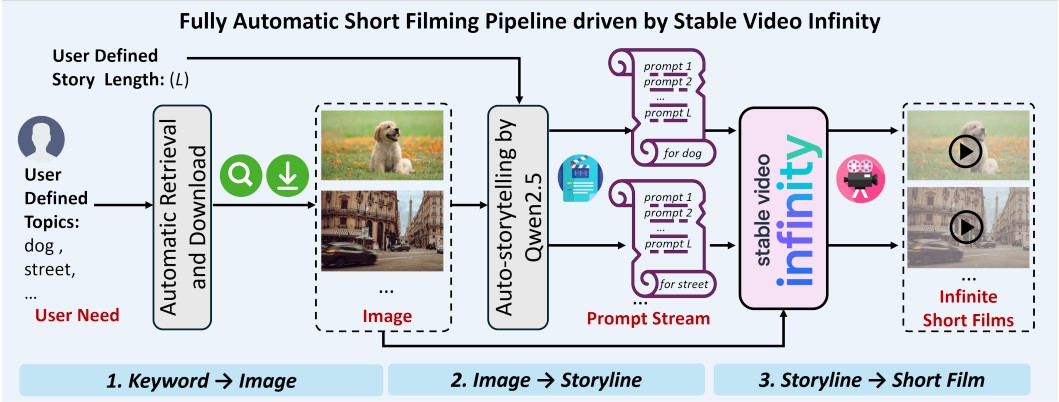

Figure 10: Overview of the proposed end-to-end automatic pipeline, which is able to generate infinite short films from user-given keywords. This engine is used to generate the prompt streams according to a specific storyline for our creative video generation benchmarks.

The complete workflow is illustrated in Fig. 10. This pipeline systematically transforms high-level keywords into structured pairs of images and prompt sequences, eliminating the need for labor-intensive manual annotation and prompt engineering. The pipeline operates as follows:

1. **Keyword-based Image Retrieval (Keyword → Image)**: The process commences with a set of user-defined keywords (e.g., "dog" and "street"). These keywords are fed into an automated download script, which retrieves a diverse set of relevant images from online resources. Optionally, the user can also skip this process by directly using the customized images instead of automatically retrieving them online.

2. **Automated Prompt Stream Generation (Image → Storyline)**: Each retrieved image is then processed by the Qwen2.5 (Yang et al., 2024a) for auto-labeling. Critically, instead of generating a single, static description, our auto-labeling module is configured to produce a temporally coherent sequence of $L$ distinct prompts. This sequence, which we term a "prompt stream", is designed to describe a plausible dynamic evolution or narrative originating from the static scene in the input image. For example, given an image of a resting dog, the prompt stream might describe the dog waking up, perking its ears, and then wagging its tail. The length of this stream, $L$, is a controllable parameter. *In our benchmark, $L$ is set to 10 for creative video generation (10 sequential video clips in one long-shot), and set to 50 for the ultra-long setting.* Optionally, users can also skip this step by directly providing their prompt streams and storyline instead of generating them via MLLM.

3. **Input Preparation and Video Synthesis (Storyline → Short Film)**: Concurrently with the prompt generation, the initially downloaded image undergoes standard normalization procedures, which are sent to SVI. The normalized image, paired with its corresponding generated prompt stream {prompt 1, prompt 2,..., prompt $L$ }, serves as the complete input to our model. *SVI will iteratively generate a video clip for each prompt within the prompt*

*stream, which uses the last frames of the previous generation as the conditions.* The model then synthesizes a video by conditioning the animation of the input image on the sequential instructions provided by the prompt stream.

This automated pipeline is central to our study. It enables rapid generation of a large and diverse suite of test cases for qualitative evaluation and provides a scalable framework for benchmarking the model's ability to interpret static content and animate it in response to dynamic textual guidance.

### A.2 BENCHMARK DATASETS

With our automatic prompt-stream engine, we are able to efficiently construct high-quality test data spanning both creative and consistent video generation scenarios. To better mirror real-world usage, two-thirds of the dataset is automatically harvested from the web through our engine, while the remaining one-third is sourced from real users following (Zhang & Agrawala, 2025), ensuring a balanced mix of scale, diversity, and authenticity. We will open-source the full codebase and all benchmark datasets to catalyze progress in long-form video generation and evaluation.

For generic video generation, we assemble 152 samples under the 50-second default setting and 14 samples under the 250-second ultra-long setting, and we evaluate all benchmarked methods under identical conditions to ensure fair comparison. In the consistent video generation track, *each input image is paired with a single, stable text prompt* across both duration settings, emphasizing temporal coherence and identity preservation. In contrast, the creative track introduces narrative dynamics: we generate *a prompt stream comprising 50 textual descriptions for the default setting and 100 for the ultra-long setting*, enabling rich, storyline-driven scene transitions and event progression. In this setting, each generated video clip (5 seconds, 16 FPS) will follow a unique prompt. Finally, for multimodal conditional generation, we evaluate 10 ultra-long samples across all benchmarked methods, probing the models' ability to align and fuse visual inputs with evolving multimodal guidance, e.g., audio and skeleton, over extended durations.

Together, these settings provide a *comprehensive, scalable, and strict benchmark* suite that stresses both fidelity and creativity, supports controlled ablations and cross-method comparisons, and reflects the practical requirements of long-video creation in the wild. This can also break through the previous long-video evaluation (Zhang & Agrawala, 2025; Henschel et al., 2025; Song et al., 2025), only focusing on a single homogeneous scene with repeated motions, which has been fully tackled by the proposed SVI-Shot, as proved by Tab. 1 in the main paper.

## B DISCUSSION

### B.1 BROADER PRACTICAL IMPACT ON INDUSTRY

**Filming and Entertainment.** Contemporary short-form video production typically requires substantial manual effort to design scene transitions, craft storylines, and stitch together video clips, making true single-take narrative videos impractical. Differently, *our Stable Video Infinity makes end-to-end, single-take filmmaking feasible and accessible.* Users provide only high-level intent and brief textual descriptions; the system then autonomously produces unlimited single-shot videos with controllable pacing and visually plausible transitions, without any human intervention.

**Robotic World Models.** Existing world models (Wang et al., 2024) for robotics (Li et al., 2025) and simulation (e.g., Cosmos (Agarwal et al., 2025)) are constrained by *short video horizons and limited training diversity, making it difficult to simulate prolonged, complex scenarios or rare out-of-distribution corner cases. Our Stable Video Infinity demonstrates strong potential for long-duration, controllable, and semantically consistent scenario synthesis, particularly in the navigation domain.*

**World Generation, Gaming, and Spatial AI.** We observe strong *long-term geometric and identity consistency across scenes* from the proposed Stable Video Infinity , a property that can catalyze progress in large-scale world generation and spatial AI. Our pipeline provides controllable, narrative-conditioned evolution of scenes while maintaining structural coherence, beneficial for research in 3D-aware video synthesis, continual scene modeling, and interactive agents that reason over persistent environments (Che et al., 2025; Yu et al., 2025; Zhu et al., 2024).

## B.2 Broader Methodology Impact on Academia

*Our central goal is to bridge the train–test hypothesis gap in autoregressive generation. This gap is pervasive across modern generative paradigms, including large language models (LLMs). and MLLM.* During training, models are typically exposed to clean, error-free inputs; at test time, however, autoregressive generation conditions each new token (or frame) on previously generated outputs, which may contain errors. Hence, this hypothesis gap is also a challenge in LLM/MLLM training. Our error-recycling tuning exposes the model to its own imperfect rollouts and teaches it to recover from errors. In doing so, the optimization becomes aligned with the test-time process.

*This principle generalizes beyond video to a broad class of generative settings, including LLMs, autoregressive image generative models.* In long video generation, the drift and compounding artifacts that destabilize content can be viewed as a form of **visual hallucination**. This has a similar property to the open problem, **linguistic hallucination** in the context of LLM and MLLM, i.e., LLM tends to give hallucinated words when the generated context is too long. Based on this insight, our error-recycling tuning potentially mitigates this effect by closing the train–test gap: the model learns robust policies for stabilization, re-grounding, and content repair under distribution shift.

## B.3 Limitation and Future Work

**Scaling Up.** Due to time constraints, our models were trained on small datasets without scaling up. We observe that when test-time image styles diverge from the training distribution, adjacent clips can exhibit color shifts. A likely cause is that the model incorrectly treats test-time low-level style as an error and "corrects" it. We plan to scale up data and diversify styles to correct this "misunderstanding", apply domain-balanced sampling, and incorporate style-preservation losses or reference-style conditioning to reduce such artifacts. Additionally, curriculum scaling, mixed/high-resolution training, and stronger augmentations should further improve robustness to style shifts.

**Real-Time and Interactive Generation.** Our current model, built on Wan 2.1, generates frames in parallel rather than as a stream, which poses challenges for real-time deployment. Recent work (e.g., CausVid (Yin et al., 2025), Self-Forcing (Huang et al., 2025)) has begun to explore streaming generation. Because our method only trains lightweight LoRA adapters, it can be seamlessly integrated into a real-time pipeline. In future work, we also plan to pursue real-time, infinite-horizon video generation and incorporate interactive control signals (e.g., live prompt updates, joystick-like trajectory guidance, and event triggers) for responsive editing and steering.

**ID Consistency.** In SVI-Film, we maintain cross-clip motion continuity by conditioning on five motion frames. However, without explicit long-term memory, when the main character exits the frame, identity drift or swapping can occur. While SVI-Shot/Talk/Dance have achieved identity control via anchor frames, they currently do not extend to creative generation with scene transitions. We intend to develop an end-to-end filming pipeline that combines persistent identity embeddings, cross-shot feature caching, and scene-aware anchors to strengthen subject consistency across complex transitions. Some advanced anchoring and memorization strategies will be proposed.

## B.4 Clarification of LLM usage

In this work, LLMs are used for writing refinement and grammatical checking, in strict accordance with ICLR guidelines. LLMs are not involved in the conception, methodology, and other sensitive components. In qualitative comparisons, we employ LLMs to assist with ancillary Python scripting tasks (e.g., vframe extraction, clip concatenation, and related utilities) and the Readme document.

## B.5 Ethical concerns

All used and benchmarked models and the corresponding training data used in this work are sourced from openly available datasets. We do not use proprietary, restricted, or sensitive data. For the retrieved test data, we have checked the permissive licenses. This study does not involve human subjects, biometric information, or biological data, and therefore does not raise associated human-subjects risks. Regarding human talking videos, we recognize potential misuse risks such as deepfakes and fraud. To mitigate these risks, future open-source releases will include compliance constraints and guardrails to discourage malicious use and support ethical deployment.

| Models | Generated Scenes | Subject Consistency | Background Consistency | Aesthetic Quality | Imaging Quality | Dynamic Degree | Motion Smoothness |
|---|---|---|---|---|---|---|---|
| Copy Clips | Single | 98.48% | 98.60% | 67.99% | 71.93% | 7.14% | 98.93% |
| Ping-Pong Clips | Single | 98.51% | 98.59% | 67.92% | 71.92% | 7.14% | 99.06% |
| Copy Reference Img | Single | 100.00% | 100.00% | 68.55% | 73.05% | 0.00% | 99.84% |
| Wan 2.1 | Single | 80.00% | 87.27% | 56.19% | 65.37% | 14.29% | 98.74% |
| StreamingT2V | Single | 66.32% | 77.62% | 40.49% | 55.18% | 85.71% | 95.60% |
| HistoryGuidance | Single | 64.84% | 80.51% | 29.84% | 50.41% | 7.14% | 99.42% |
| FramePack | Single | 79.37% | 86.64% | 55.66% | 57.61% | 0.00% | 99.63% |
| SVI-Shot (Ours) | Single | 97.50% | 97.89% | 65.75% | 71.54% | 21.43% | 98.81% |

Table 5: Exploring naive video extension methods. The best is highlighted in red (abnormally large).

| Error | Sub. Cons. | Back. Cons. | Aest. Qual. | Img. Qual. |
|---|---|---|---|---|
| Self Only | **69.34%** | 83.14% | **52.83%** | **56.97%** |
| Handcraft Only | 69.21% | **83.65%** | 49.62% | 45.17% |
| Self+Handcraft | 69.24% | 83.48% | 48.51% | 39.50% |

Table 6: Comparison between self-generated errors and handcraft errors with image augmentation.

| $\alpha$ | Sub. Cons. | Back. Cons. | Aest. Qual. | Img. Qual. |
|---|---|---|---|---|
| 0.2 | 92.16% | 93.68% | 58.52% | 77.49% |
| 0.4 | 97.49% | 96.73% | 59.86% | 77.32% |
| 0.8 | **99.59%** | **99.37%** | 59.69% | 78.11% |
| 1.0 | 98.54% | 97.97% | **59.88%** | **78.12%** |

Table 7: Analysis on error-recycling intensity by modifying LoRA alpha.

## C  QUANTITATIVE EXPERIMENTS

**Exploring Naive Ways of Fooling Metrics.** In Tab 5, we further study three types of designs fooling metrics: *(a) Copy Clips*, which copies the first generated video clips 50 times naively; *(b) Ping-Pong Clips* copies the first generated video clips 50 times in a ping-pong manner, and *(c) Copy Reference Image* only naively copies reference images by $50 \times 81$ times. We can see that these naive methods can successfully fool and attack some metrics, e.g., consistency and quality, showing some limitations of existing video generation evaluation. Correspondingly, methods with abnormally high values on these metrics tend to exhibit abnormally low counterparts, for example, a 0.00% dynamic degree. *Hence, we can learn a valuable message that it is necessary to comprehensively consider all metrics together for evaluation to prevent metric fooling.* Our SVI gives a satisfactory trade-off among all metrics, revealing its effectiveness.

**Comparison between Self-generated Error and Naive Image Augmentation.** In Tab. 6, we apply handcrafted degradations to the reference image, including random color shifts, blur, and sharpness, and compare with our self-generated errors. We observe that the naive image augmentations not only fail to help but also substantially degrade image quality. Moreover, combining self-generated and handcrafted errors causes severe conflicts, leading to further drops. These results suggest that accumulated, model-induced errors possess unique characteristics that are difficult to mimic with manual augmentations, showing the necessity of learning from the model's own errors.

**Analysis on Error-recycling Intensity.** In Tab. 7, bottom, we gradually adjust the error-recycling intensity by changing LoRA weight $\alpha$ in test, where a lower value means a weaker effect. Compared with $\alpha = 1$, there is a consistent decrease in all metrics when reduced from $0.8$ to $0.2$, indicating that the more catastrophic errors appear when weakening the error correction ability. Hence, this can justify the necessary role of correcting errors actively.

**Analysis on Error Bank Size.** Tab. 8 evaluates the effect of varying the error bank size $Z$ on model performance across multiple metrics. Exceedingly limited error bank sizes, such as $Z = 1$ or $Z = 10$, restrict error diversity, leading to suboptimal performance in Subjective Consistency, Background Consistency, Aesthetic Quality, and Image Quality. As $Z$ increases, all metrics show consis-

| $Z$ | Sub. Cons. | Back. Cons. | Aest. Qual. | Img. Qual. |
|---|---|---|---|---|
| 1 | 67.82% | 82.90% | 51.55% | 52.96% |
| 10 | 67.41% | 81.79% | 52.12% | 54.29% |
| 100 | 68.51% | 82.97% | 51.03% | 55.30% |
| 500 | **69.34%** | 83.14% | **52.83%** | **56.97%** |
| 1000 | 69.14% | **83.83%** | 51.36% | 55.06% |
| 2000 | 69.02% | 82.98% | 51.39% | 54.55% |

Table 8: Ablation study on error bank size $Z$.

tent improvements. However, beyond $Z = 500$, performance saturates, with most metrics exhibiting no significant gains or slight declines. Our selection of $Z = 500$ achieves satisfactory performance, effectively balancing error diversity with model capability.

| LoRA Rank | Subject Consistency | Background Consistency | Aesthetic Quality | Imaging Quality | Dynamic Degree | Motion Smoothness |
|---|---|---|---|---|---|---|
| 0 (baseline) | 66.23% | 81.96% | 45.55% | 44.29% | **62.50%** | 98.12% |
| 64 | 95.37% | 95.76% | 62.17% | 70.34% | 31.25% | 98.95% |
| 128 | **96.10%** | **96.75%** | **62.84%** | **70.62%** | 25.00% | **99.12%** |

Table 9: Exploring different LoRA ranks, where larger ranks indicate more trainable parameters.

| Models | Error Injection | Subject Consistency | Background Consistency | Aesthetic Quality | Imaging Quality | Dynamic Degree | Motion Smoothness |
|---|---|---|---|---|---|---|---|
| Wan 2.1 | No Error | 81.44% | 89.81% | 51.33% | 53.09% | **61.97%** | 98.57% |
| SVI | Offline | 93.52% | 95.86% | 58.07% | 62.81% | 55.63% | 98.42% |
| SVI | Online: Single-pass | 94.65% | 95.65% | 58.66% | 64.85% | 47.74% | 98.92% |
| SVI | Online: Two-pass | **94.96%** | **95.89%** | **59.31%** | **65.24%** | 35.48% | **99.08%** |

Table 10: Exploring different online error injection variations.

**The Sufficiency of LoRA Tuning.** In Tab. 9, to further study the required capacity of LoRA, we conduct studies on the LoRA rank, where the larger rank indicates more trainable parameters with larger capacity. Compared to the baseline, even a small-parameter version with rank 64 already yields a clear performance gain. Moreover, when we increase the rank from our 64 to 128, the improvement is marginal, suggesting that LoRA has sufficient capacity to remove self-errors.

**Exploring Online Error Injection.** In Tab. 10, we further compare three error-injection variations. (1) *Offline injection.* The injected error is drawn from the corresponding memory bank, as in our previous design. (2) *Single-pass online injection.* For each clean sample, we first forward-propagate once to estimate the error and compute the gradient for optimization. Next, we inject this sample's own error and perform another gradient update using the error-injected sample. (3) *Two-pass online error injection.* Similar to the single-pass setting, we first run the sample through the model to estimate the first-round error. We then inject this error, perform forward propagation, and re-compute the accumulated error. Finally, we inject the second-round accumulated error and perform the gradient update using this cumulatively error-injected sample. We can find that these online variants slightly improve most metrics compared to the original SVI, except for the dynamic-degree metric. We hypothesize that this is because online error injection enhances restoration capacity by better matching samples to errors, thereby improving consistency and quality. However, it may also reduce the diversity of sample–error pairs, thereby decreasing motion.

**Evaluation via Existing Benchmarks.** In Tab. 11, We further evaluate SVI via VBench benchmark (Huang et al., 2024a) (both images and prompts). Due to the high computational cost of evaluating long videos, we randomly sample 50 test cases and assess performance at 50-clip lengths, as in our main paper. The results show that SVI consistently outperforms other methods on the standard benchmark, verifying its effectiveness and generalization ability.

# D  IMPLEMENTATION DETAILS

The experiments are conducted on a large-scale GH200 cluster. Detailed hyperparameters used for SVI training are shown in Tab. 12. We implement SVI based on Wan2.1-I2V-14B-480P, and only tune LoRA to enable the flexibility, i.e., the user can effortlessly inject SVI into their private models. **All the models/source codes/benchmark datasets will be made publically available.**

**Training Data.** The proposed Stable Video Infinity is significantly data efficient as it only uses small-scale publicly available data to fine-tune. In all settings, we only train SVI with 10 epochs. For the creative and consistent video generation setting, the proposed SVI-Shot and SVI-Film are trained with the MixKit Dataset (Lin et al., 2024) consisting of 6K videos. For the audio-guided talking. we use a random subset of Hallo 3 (Cui et al., 2025) containing 5,000 video clips for training. For the skeleton conditional dancing, we use TikTok (Jafarian & Park, 2021) for the error-recycling fine-tuning, where the LoRA is pretrained from (Wang et al., 2025b).

| Models | Subject Consistency | Background Consistency | Aesthetic Quality | Imaging Quality | Dynamic Degree | Motion Smoothness |
|---|---|---|---|---|---|---|
| Wan 2.1 | 76.11% | 86.56% | 56.19% | 63.66% | **54.00%** | 98.37% |
| HistoryGuidance | 63.60% | 82.10% | 27.80% | 48.70% | 8.00% | 99.37% |
| FramePack | 78.11% | 86.13% | 54.64% | 55.57% | 0.00% | **99.49%** |
| SVI (Ours) | **96.24%** | **96.13%** | **64.36%** | **68.54%** | 30.00% | 98.75% |

Table 11: VBench benchmark results for long-video generation (50-clip).

| Parameter | Value | Description |
|---|---|---|
| Learning rate | 2.0e-05 | Adam optimizer learning rate |
| Max epochs | 10 | Maximum training epochs |
| Gradient clipping | 1.00 | Gradient norm clipping threshold |
| Gradient accumulation | 1 | Gradient accumulation steps |
| Training strategy | deepspeed_stage_2 | Distributed training |
| Data workers | 1 | Number of data loading workers |
| Gradient checkpointing | Yes | Memory optimization technique |
| Checkpointing offload | Yes | CPU gradient checkpointing |
| LoRA rank | 128 | Low-rank adaptation rank dimension |
| LoRA alpha | 128 | LoRA scaling parameter |
| LoRA init | kaiming | LoRA weight initialization method |
| Architecture | lora | Training architecture type |
| LoRA position | q,k,v,o,ffn.0,ffn.2 | Target modules for LoRA |
| Frame height | 480 | Video frame height (pixels) |
| Frame width | 832 | Video frame width (pixels) |
| Tiled processing | Yes | Tiled inference for memory efficiency |
| Tile height | 34 | Processing tile height |
| Tile width | 34 | Processing tile width |
| Video frames | 81 | Number of video frames per sample |
| error-recycling tuning | Yes | Enable error-recycling tuning |
| Warmup iterations | 20 | Number of iter. gathering multi-node errors |
| Noise error $p_{\mathrm{noi}}$ | 0.01 | Noise error injection probability |
| Latent error $p_{\mathrm{vid}}$ | 0.9 | Latent error injection probability |
| Image error $p_{\mathrm{img}}$ | 0.9 | Image error injection probability |
| Clean input $p$ | 0.5 | probability without any error |
| Timestep grids | 50 | Discretized timestep grids for error buffer |
| Maximum error $Z$ | 500 | Maximum errors saved in each memory grid |
| Motion grames | 5 | Number of motion reference frames |
| Motion probability | 0.95 | Probability of using motion frame |

Table 12: Detailed hyperparameters used in the training and test.

# E   ADDITIONAL QUALITATIVE COMPARISON

The proposed SVI demonstrates capability in generating temporally coherent short films guided by text streams, as evidenced in Fig. 11 through Fig. 15, showcasing its potential for end-to-end storytelling and creative content creation applications. Beyond basic text-to-video generation, our method exhibits remarkable versatility by supporting multimodal controls. As illustrated in Fig. 16 and Fig. 17, SVI achieves robust long-range video synthesis through both visual conditioning and embedding-based control, enabling precise manipulation of character movements and facial expressions. Please refer to the anonymous project page https://anonymous.4open.science/w/Stable-Video-Infitity-51DE/ for the video comparison.

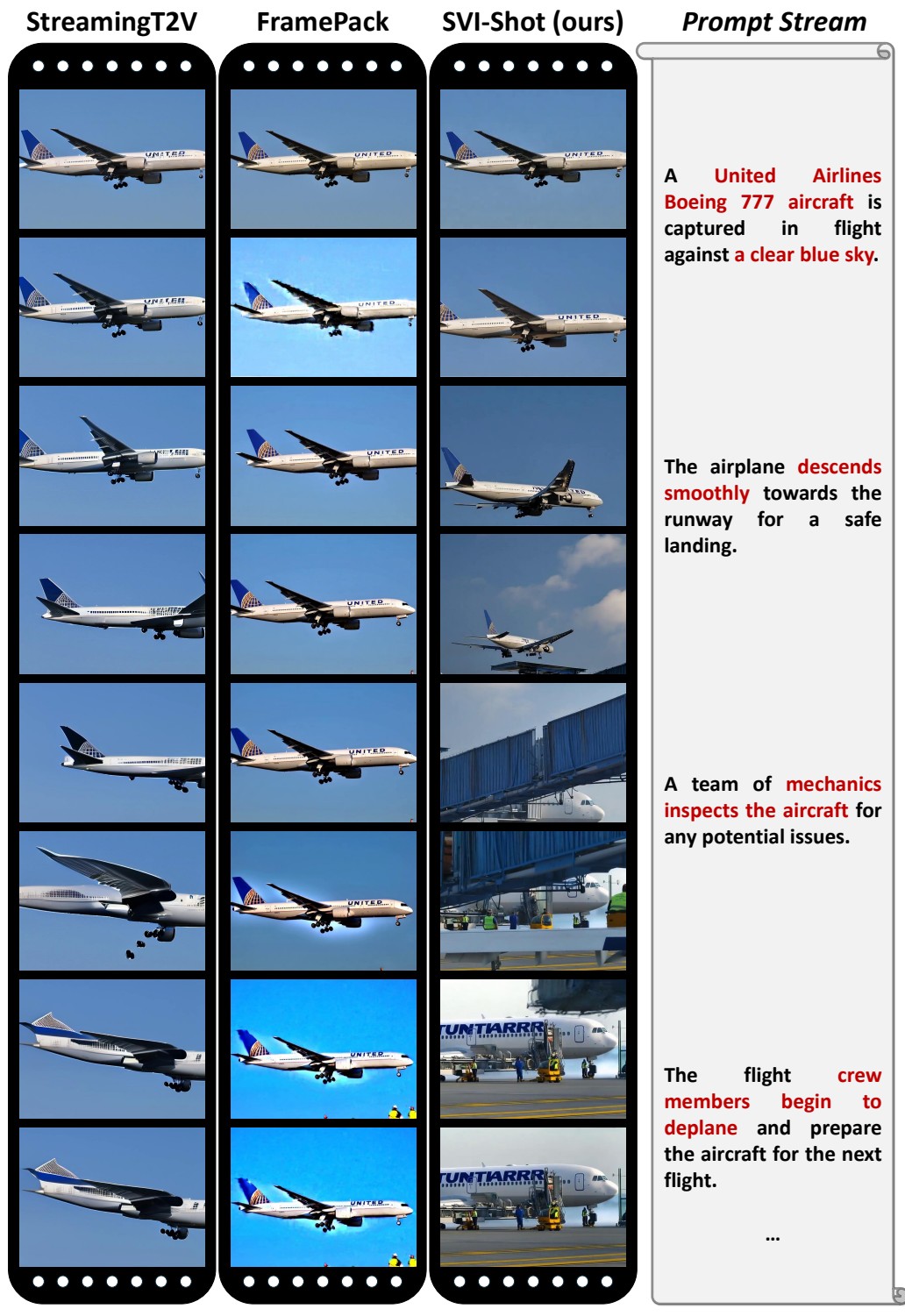

Figure 11: Qualitative results about airplane landing story.

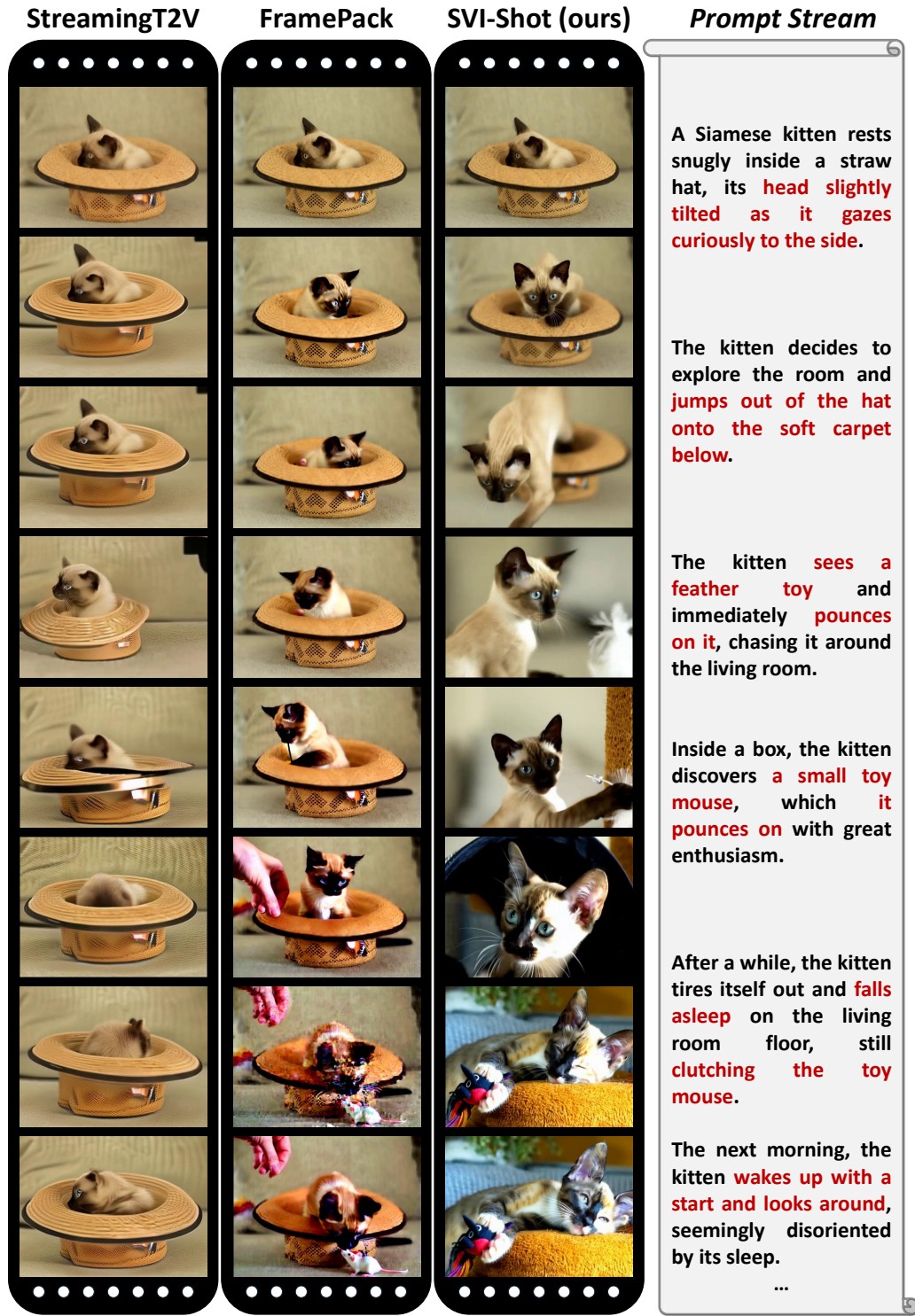

Figure 12: Qualitative results about the cat story.

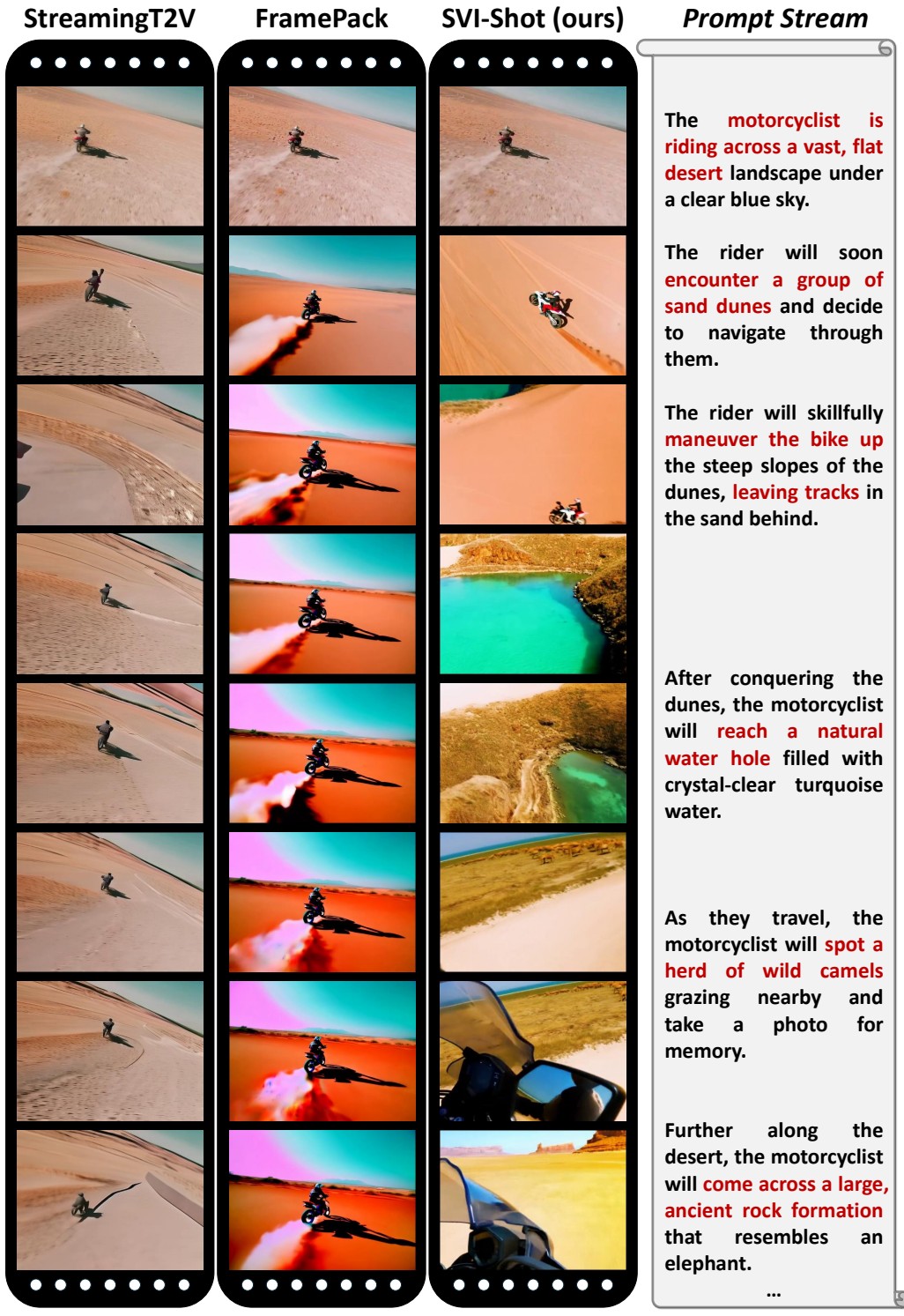

Figure 13: Qualitative results about the motorcycle story.

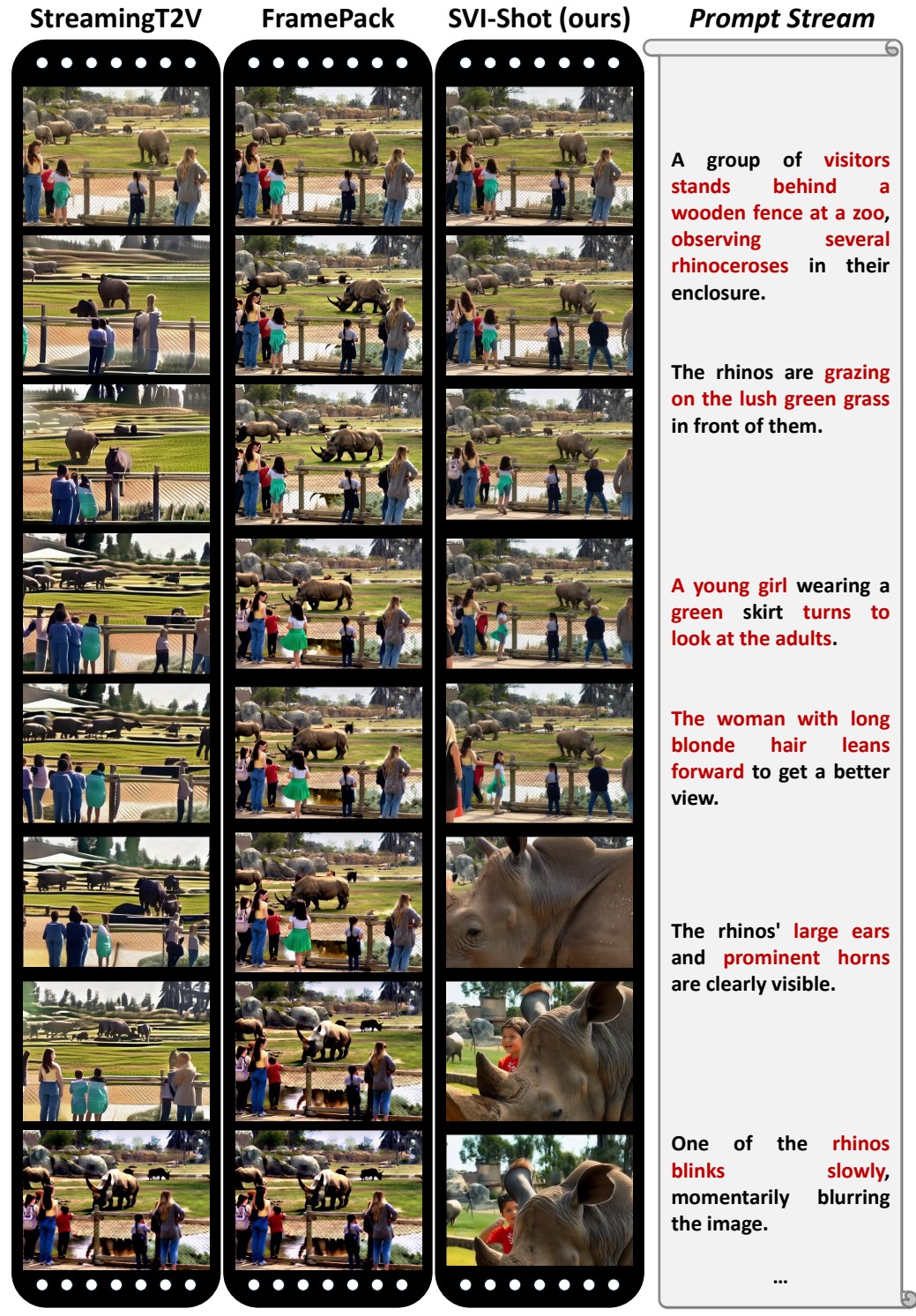

Figure 14: Qualitative results about the zoo story.

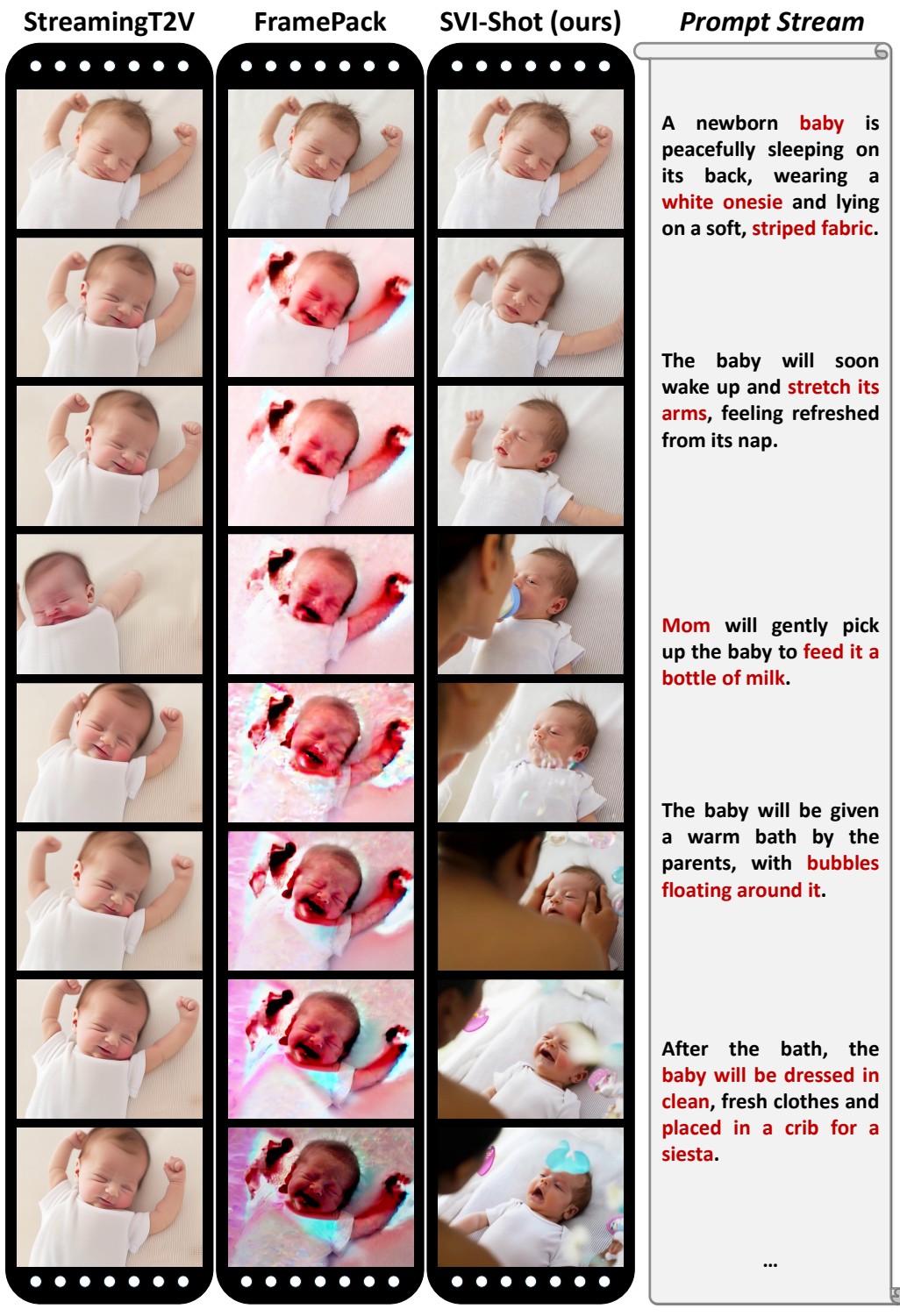

Figure 15: Qualitative results about the baby story.

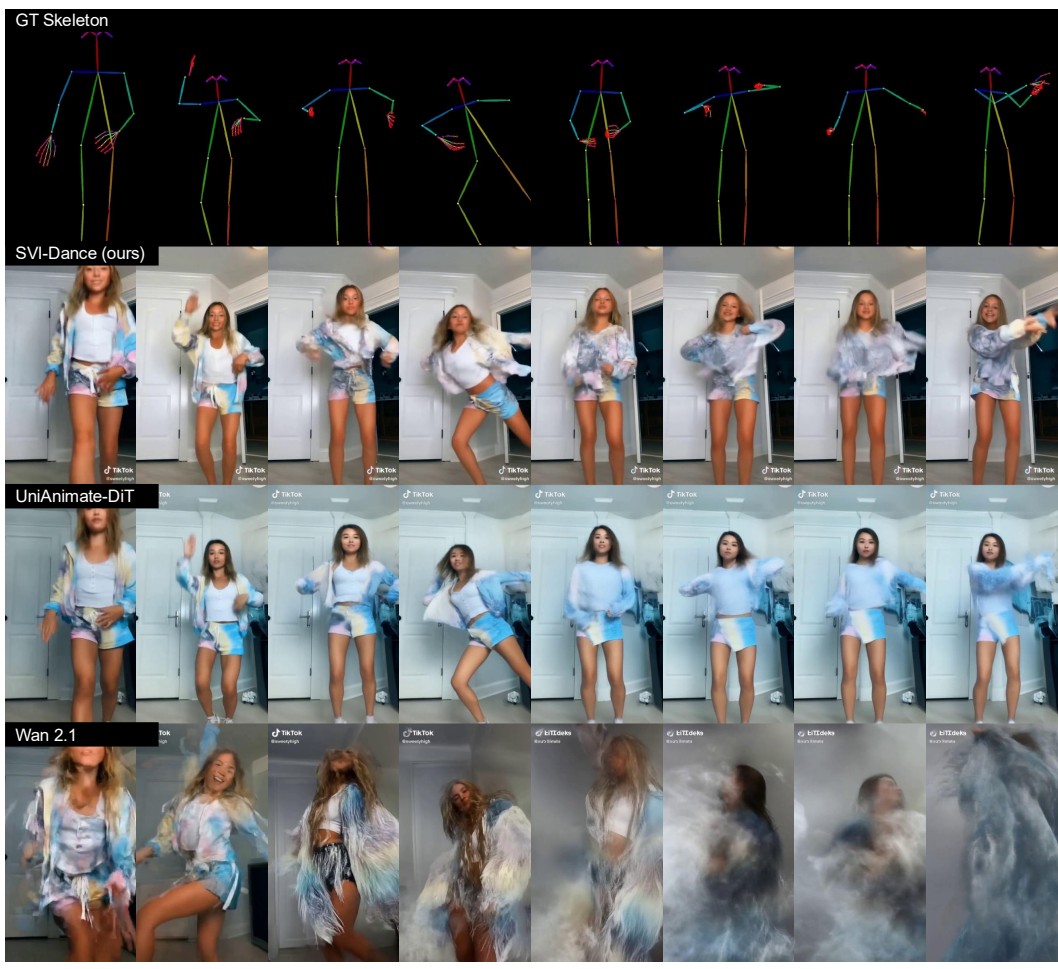

Figure 16: Qualitative results about the dancing.

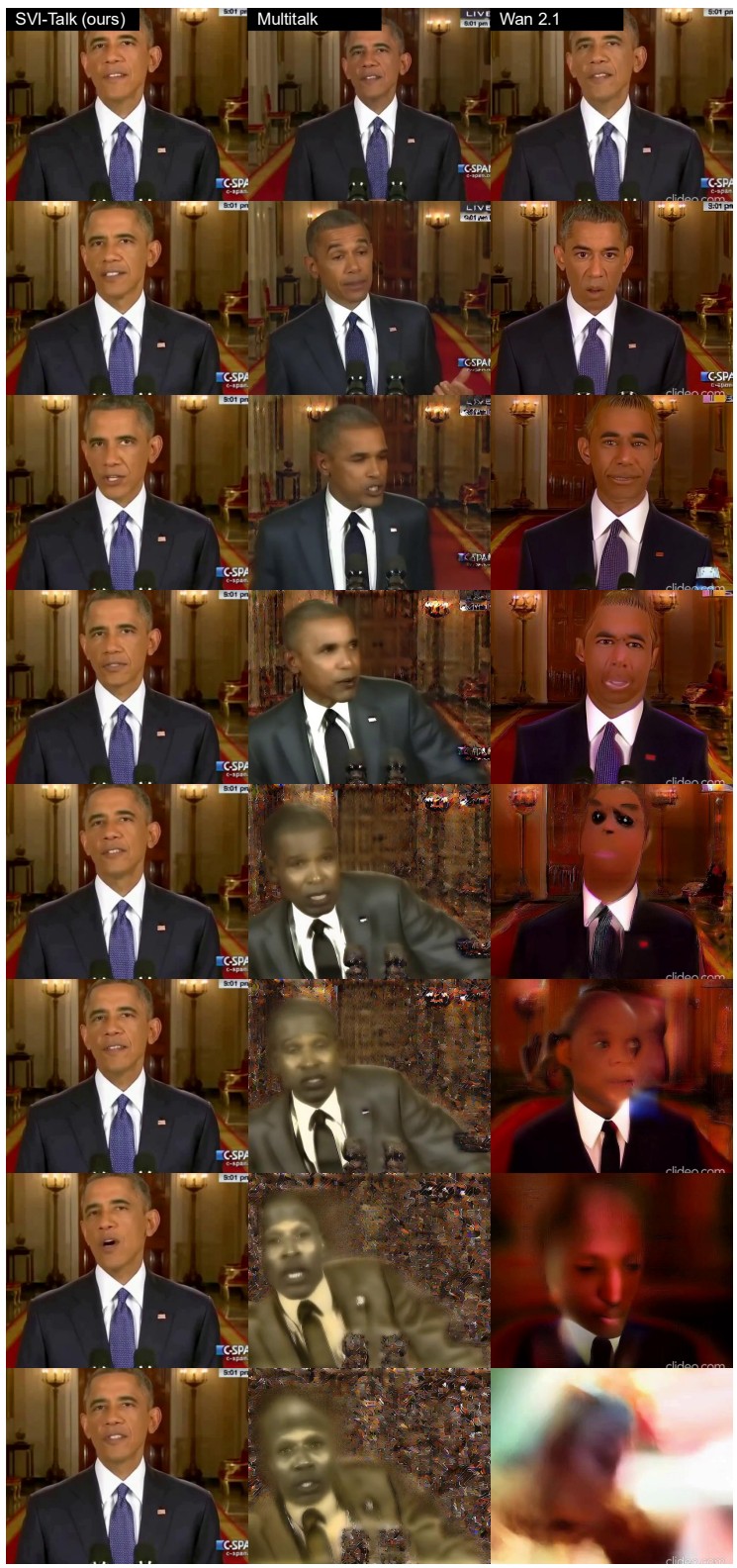

Figure 17: Qualitative results about the talking face.

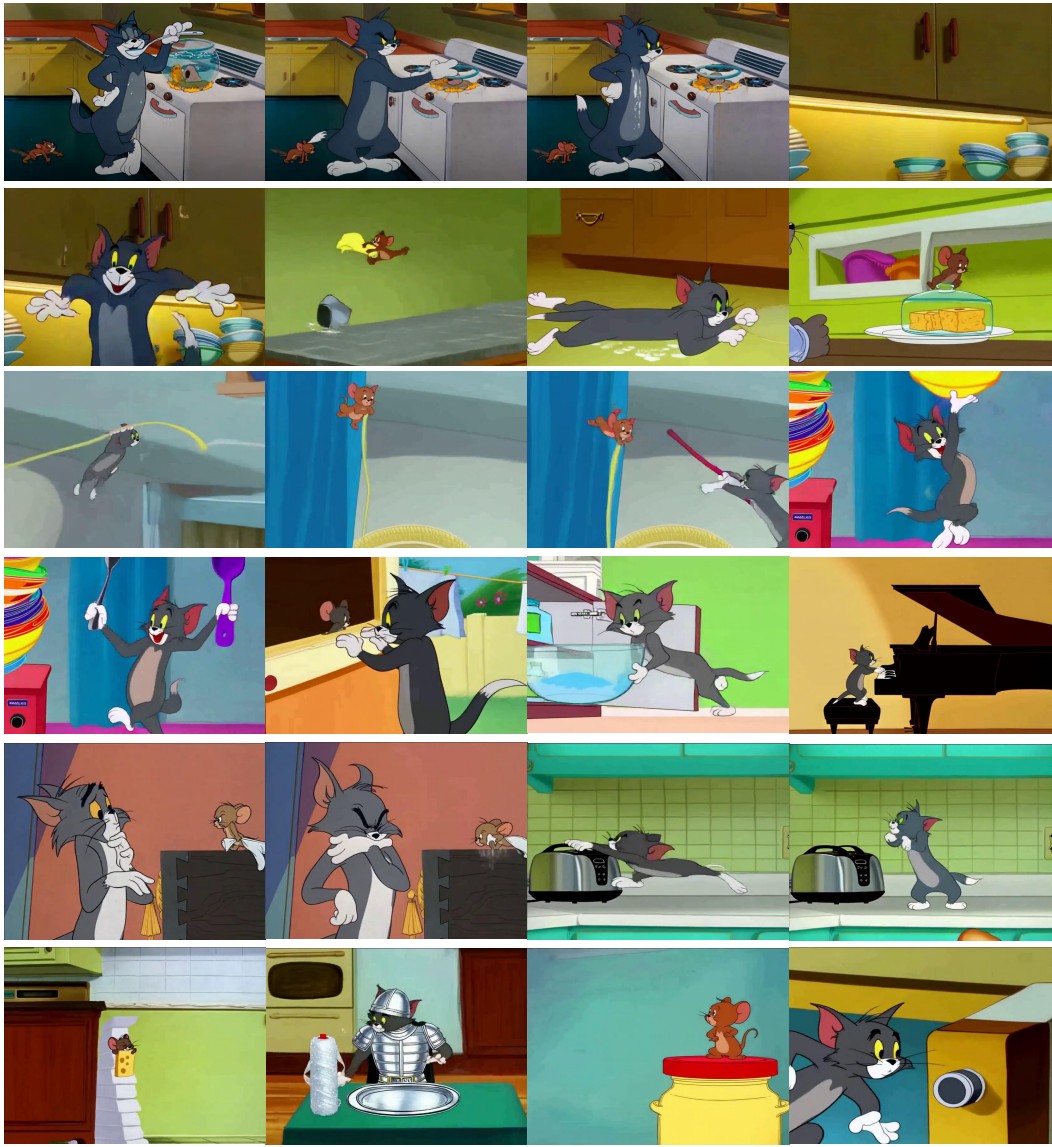

Figure 18: Qualitative results about the clips of Tom and Jerry.

