# OpenReview forum: "Stable Video Infinity: Infinite-Length Video Generation with Error Recycling"
_ICLR.cc/2026/Conference — ICLR 2026 Oral_

### Official Review · Reviewer_rurm · 2025-10-24

**Soundness:** 3
**Presentation:** 3
**Contribution:** 3
**Rating:** 8
**Confidence:** 5

**Summary:**

This paper presents Stable Video Infinity, which can generate infinite long video without quality degradation. It regards the long-video generation problem as a train-test hypothesis gap issue rather than a mere accumulation of generation error. During training, models are fed clean ground-truth latents and reference frames; at inference, to generate long video, they have to autoregressively condition on their own previously-generated, error-containing outputs. This mismatch brings two coupled error modes: single-clip predictive error and cross-clip conditional error. Instead of engineering stronger noise-schedulers or anchor frames, the proposed method recycle the model’s own error back into the training to avoid quality drop during generation. I generally like this work and only have minor questions.

**Strengths:**

1. The idea of considering the training and inference gap is interesting. Unlike previous methods that generally considering heuristic drift mitigation, SVI aims to loop the generated error into training to reduce the gap during testing.

2. A Comprehensive analysis on the accumulated error in video generation is provided.

3. Comprehensive experiments with various setting are conducted. The proposed method achieve higher results compared to other competitors.

**Weaknesses:**

1. While injecting the error is interesting, in implementation it is not the online error injecting, which means the injected error is exactly the generated one, but instead an error sampled from the bank. This may cause some theoretical issues.

2. The memory bank is capped at 500. However, the bank is filled by federated gathering across GPUs; if each GPU sees only a very few of videos, the effective diversity may be far smaller than 500, leading to the homogenization of errors.

**Questions:**

Please see the questions in weaknesses.

---

> ### Author Response · Authors · 2025-11-19
>
> We sincerely appreciate your thoughtful review and positive evaluation of our work. We hope that our detailed, point-by-point replies have adequately addressed your concerns.
>
> ## [W1] Exploring the online error injection variations
>
> We really appreciate your insightful and constructive suggestion. Following this advice, we further experimented with and compared three error-injection variations:
>
> 1. **Offline injection.** The injected error is drawn from the corresponding memory bank, as in our previous design.
>
> 2. **Single-pass online injection.** For each clean sample, we first forward-propagate once to estimate the error and compute the gradient for optimization. Next, we inject this sample's own error and perform another gradient update using the error-injected sample.
>
> 3. **Two-pass online error injection.** Similar to the single-pass setting, we first run the sample through the model to estimate the first-round error. We then inject this error, perform forward propagation, and recompute the accumulated error. Finally, we inject the second-round accumulated error and perform the gradient update using this cumulatively error-injected sample.
>
> The experimental results are shown in the following table. We are surprised to find that these online variants slightly improve most metrics compared to the original SVI, except for the dynamic-degree metric. We hypothesize that this is because online error injection enhances restoration capacity by better matching samples to errors, resulting in improved consistency and quality. However, it may also reduce the diversity of sample-error pairs, thereby decreasing motion. We have updated this experiment in the revised manuscript (Table 10; Page 18).
>
> | Models   | Error  Injection      | Subject  Consistency | Background  Consistency | Aesthetic  Quality | Imaging  Quality | Dynamic  Degree | Motion  Smoothness |
> |----------|-------------------------|------------------------|---------------------------|----------------------|--------------------|-------------------|----------------------|
> | Wan 2.1  | No Error                | 81.44%                 | 89.81%                    | 51.33%               | 53.09%             | **61.97%**        | 98.57%               |
> | SVI      | Offline                 | 93.52%                 | 95.86%                    | 58.07%               | 62.81%             | 55.63%            | 98.42%               |
> | SVI      | Online: Single-pass     | 94.65%                 | 95.65%                    | 58.66%               | 64.85%             | 47.74%            | 98.92%               |
> | SVI      | Online: Two-pass        | **94.96%**             | **95.89%**                | **59.31%**           | **65.24%**         | 35.48%            | **99.08%**           |
>
> ## [W2] Clarifying the error bank diversity
>
> Thank you for your insightful comments! Actually, we have also considered this concern during algorithm design and tried to explicitly alleviate the potential "error homogenization" problem from two perspectives.
>
> 1. We explicitly encourage both a sufficient number and a sufficient variety of errors. The federated gathering stage is designed to guarantee quantity: with a global batch size of 64, every 100 training iterations can generate about 6,400 distinct error instances. This quickly fills the error memory and provides enough error samples for optimization on each GPU. To maintain diversity, we also apply a dynamic memory update strategy and adopt random error sampling, thereby enhancing the error distribution.
>
> 2. We have also explored additional techniques to manually increase error diversity, such as random magnitude modulation and data augmentation of errors. However, we did not observe clear performance gains from these variants. We hypothesize that, under the current design, the diversity of the collected errors is already sufficient for effective learning and fast model convergence.

---

> > ### Comment · Reviewer_rurm · 2025-11-25
> >
> > Thank you for the clarification. I will keep my score.

---

> > > ### Author Response · Authors · 2025-11-25
> > >
> > > Dear Reviewer rurm,
> > >
> > > We are very grateful for your constructive feedback and for maintaining your positive rating. We also sincerely appreciate the time and effort you have devoted to reviewing our work.
> > >
> > > Best,
> > > Authors of Submission #2020

---

### Official Review · Reviewer_8N9r · 2025-10-27

**Soundness:** 2
**Presentation:** 3
**Contribution:** 3
**Rating:** 6
**Confidence:** 3

**Summary:**

The paper addresses the fundamental challenge in generating very long videos: the discrepancy between error-free training with clean data and error-accumulating inference, which autoregressively operates on self-outputs with increasing error accumulation. They propose Error-Recycling Fine-tuning, which integrates the historical errors (degraded input) into training, to minimize the train-test gap. SVI deliberately injects historical errors into clean inputs and learns to predict an error-recycled velocity. THeir method achieves strong performance relative to baselines on their video generation benchmark.

**Strengths:**

1. The paper motivates well the core challenge for long video generation, i.e., that errors accumulate from self-outputs, and their proposed solution, i.e., put self-predicted errors into training.
2. Their proposed method achieves strong performance relative to baselines on their video generation benchmark.
3. The ablation study for each error term was helpful to empirically demonstrate that the image error term was the most important piece of their training strategy.

**Weaknesses:**

1. I think the paper would benefit from hedging its claims a little more. For example, they say "we propose Stable Video Infinity (SVI) that can generate infinite-length videos with temporally coherent and visually plausible context following a long storyline." This seems like a stretch considering that the evaluations are on the order of a few minutes, and it is challenging to evaluate the storyline.
2. The paper would benefit from comparing how long it takes for training and inference for their method relative to baseline methods. Does the proposed method take longer to train? Does this method take longer for inference?
3. The paper develops their own benchmark (which is well-motivated), but it would also be helpful to compare their proposed method against baselines on an existing benchmark (or a subset of one, if computationally expensive).

**Questions:**

1. There are other papers with a similar idea of addressing this fundamental train-test prediction error gap in long video generation by training models with their own outputs/errors. These papers are relatively recent, but it might still be beneficial to discuss: how is the proposed method (conceptually) similar to and different from these other works? Here are some examples:
- Self Forcing: Bridging the Train-Test Gap in Autoregressive Video Diffusion
- Rolling Forcing: Autoregressive Long Video Diffusion in Real Time
- others?
2. In Figure 5, why do some metrics show a decrease then increase?

---

> ### Author Response · Authors · 2025-11-19
> **Author Response to Weakness 1-3**
>
> We really appreciate your professional review! We hope that our point-to-point responses can address your concerns.
>
> ## [W1] Hedging the claim
>
> Thank you very much for this suggestion. We fully agree and have refined our claim as follows:
>
> SVI can generate non-looping, infinite-length videos with stable visual quality, while supporting per-clip prompt control and multi-modal conditioning.
>
>
> ## [W2] Clarifying training and inference efficiency
>
> SVI only trains LoRA on a small dataset, making training highly efficient and imposing no extra computational burden at inference time. Since the compared methods, e.g., FramePack, are full-model-tuned (not LoRA-based) and do not release their training code or data, it is difficult to compare their absolute training time fairly. Hence, we provide a rough comparison. Compared to FramePack, which requires 48 hours of training, SVI can be trained with 8 GPUs for about one day (24 hours) to achieve satisfactory performance. For our final model, we use 64 GPUs to train 10 epochs, which takes roughly 10 hours.
>
> At inference time, SVI has the same runtime as the underlying backbone (Wan 2.1). Generating a 5-second clip takes about 7 minutes with the standard Wan 2.1 setup on one H100 GPU. With step distillation LoRA from the LightX2V Github repository [1], the inference time can be further reduced to around 120 seconds for a 5-second clip on a commercial NVIDIA 4090 GPU, which is similar to Teacache-accelerated FramePack (around 120 seconds).
>
>
>
> ## [W3] Evaluations under existing benchmarks
>
> Thank you for your suggestion. We further evaluated SVI using the official VBench samples (both images and prompts). Due to the high computational cost of long-video evaluation, we randomly sampled 50 test cases and assessed performance with 50-clip lengths, consistent with our main paper. The results show that SVI consistently outperforms other methods on this standard benchmark, demonstrating its effectiveness and generalization ability. We have updated the experiments in the revised manuscript (Table 11, Page 19).
>
> | Models           | Subject  Consistency | Background  Consistency | Aesthetic  Quality | Imaging  Quality | Dynamic  Degree | Motion  Smoothness |
> |------------------|------------------------|---------------------------|----------------------|--------------------|-------------------|----------------------|
> | Wan 2.1          | 76.11%                 | 86.56%                    | 56.19%               | 63.66%             | **54.00%**        | 98.37%               |
> | HistoryGuidance  | 63.60%                 | 82.10%                    | 27.80%               | 48.70%             | 8.00%             | 99.37%           |
> | FramePack        | 78.11%                 | 86.13%                    | 54.64%               | 55.57%             | 0.00%             | **99.49%**               |
> | SVI (Ours)       | **96.24%**             | **96.13%**                | **64.36%**           | **68.54%**         | 30.00%            | 98.75%               |

---

> > ### Author Response · Authors · 2025-11-19
> > **Author Response to Question 1-2**
> >
> > ## [Q1] Discussing the recent related works
> >
> > We appreciate the reviewer pointing out these recent and concurrent works, and we have updated the corresponding discussion in the revised manuscript (Sec. 6; Page 10). Conceptually, all these methods and ours share a similar high-level goal: reducing the mismatch between training on clean contexts and inference under autoregressive, error-prone histories. Differently, instead of constraining self-generated rollouts, SVI explicitly targets inference errors, i.e., the essential cause of the train-test gap, addressing it from a new error-recycling perspective.
> >
> > Specifically, **Self-Forcing** [2] bridges the gap by performing autoregressive rollouts with KV cache during training and using distribution-matching on self-generated videos. Built upon this, **Rolling Forcing** [3] solves the long-video streaming problem with rolling-window joint denoising and attention sink to suppress error accumulation. Similarly, **LongLive** [4] further specializes in interactive, prompt-driven long videos by combining frame-level AR with streaming long tuning, KV re-cache for prompt switching, and short-window attention plus frame-sink to balance long-range consistency. In contrast to restructuring rollout videos, **SVI** explicitly focuses on inference errors (the essential cause of drifting) and encourages models to eliminate their own prediction/conditioning errors via error-recycling. This is realized as a more lightweight fine-tuning with two unique advantages: (i) data- and compute-efficient, and (ii) easily extensible to diverse conditions (e.g., text streams, audio, skeleton control) and scene transitions for ultra-long videos. In this sense, works based on Self-Forcing are well-suited for real-time streaming and interaction, while our approach provides a lightweight, flexible, and multi-conditional solution.
> >
> >
> > ## [Q2] Why do some metrics decrease and then increase?
> >
> > We have also observed this interesting and somewhat unique “drop-then-rise” behavior in the metrics, and we believe it stems from two main factors:
> >
> > 1. **The reason for the first drop.** Each clip is generated with some randomness (e.g., the random quality of the last frame in the previous clip and the random seed of the current clip); a few outlier clips are likely to appear after multiple generations. This can occur even for standard 5-second short video generation and may lead to slight local degradation in visual quality or consistency, resulting in an initial drop in the metrics. We have also updated an extremely ultra-long video (19-minute) on the original anonymous project page (the link is at the end of the abstract). Here, we can find some lower-quality generation in some middle clips, which can be corrected by the model in subsequent generations, revealing the error correction ability.
> >
> > 2. **The reason for the later rise.** Thanks to SVI's error-correction nature, the model tends to detect these outlier clips and gradually correct them in subsequent clips, improving visual quality. In addition, as the video becomes longer, the relative influence of a few outlier clips on the global metrics can also be smoothed over time.
> >
> > ## Reference List
> >
> > [1] LightX2V: Light Video Generation Inference Framework
> > [2] Huang, X., Li, Z., He, G., Zhou, M., & Shechtman, E. (2025). Self Forcing: Bridging the Train-Test Gap in Autoregressive Video Diffusion. arXiv preprint arXiv:2506.08009.
> > [3] Liu, K., Hu, W., Xu, J., Shan, Y., & Lu, S. (2025). Rolling Forcing: Autoregressive Long Video Diffusion in Real Time. arXiv preprint arXiv:2509.25161.
> > [4] Yang, S., Huang, W., Chu, R., Xiao, Y., Zhao, Y., Wang, X., ... & Chen, Y. (2025). Longlive: Real-time interactive long video generation. arXiv preprint arXiv:2509.22622.

---

> > > ### Comment · Reviewer_8N9r · 2025-11-22
> > >
> > > [W1] Hedging the claim
> > > - I think one part of the claim left to be hedged is "infinite-length". As I mentioned in my comment "This seems like a stretch considering that the evaluations are on the order of a few minutes". The authors proposed a new statement "SVI can generate non-looping, infinite-length videos with stable visual quality". I think the "infinite-length" part is not proven. It's fine to say this sentence because it is the method is theoretically infinite-length, but it is not empirically shown. But overall I think this is minor.
> > >
> > > [W2, W3, Q1, Q2] I am happy with the author's reply. I think this improves the paper, and I recommend they integrate parts of their rebuttal into the main paper.
> > >
> > > I will improve my review score.

---

> > > > ### Author Response · Authors · 2025-11-22
> > > >
> > > > Dear Reviewer 8N9r,
> > > >
> > > > We sincerely appreciate your careful and rigorous review, and we are very glad that we were able to address your concerns. We agree with your comment regarding the phrasing of "infinite-length", and we appreciate your professional perspective on this point. To better align our claim with the scope of our experiments and benchmark design, we will further hedge it to: *SVI can generate non-looping, ultra-long videos with stable visual quality*.
> > > >
> > > > Thank you again for your constructive feedback, which has greatly helped us improve the quality of our paper.
> > > >
> > > > Best,
> > > > Authors of Submission #2020

---

### Official Review · Reviewer_ci1H · 2025-10-31

**Soundness:** 2
**Presentation:** 3
**Contribution:** 2
**Rating:** 4
**Confidence:** 4

**Summary:**

This paper proposes Stable Video Infinity, a novel framework for infinite-length video generation using a Diffusion Transformer (DiT). The core contribution is an Error-Recycling Fine-Tuning (ERFT) strategy that addresses the train–test hypothesis gap in long video generation. Unlike previous anti-drifting or noise-rescheduling approaches that merely alleviate accumulated prediction errors, SVI explicitly recycles self-generated errors as supervisory prompts. The method injects historical errors into clean inputs, approximates error trajectories via one-step bidirectional integration, stores them in replay memory, and selectively resamples them for future timesteps. This closed-loop tuning enables DiTs to actively correct their own mistakes, scaling short-video models to arbitrarily long sequences. Extensive experiments across consistent, creative, and conditional benchmarks demonstrate superior temporal stability, scene diversity, and multimodal controllability over prior methods such as FramePack, StreamingT2V, and HistoryGuidance.

**Strengths:**

1. The research problem of error accumulation in long video generation is crucial, and the proposed pipeline effectively addresses it.
2. The experiments cover various scenarios (long videos, creative storytelling, multimodal control), providing both quantitative and qualitative results with well-defined baselines.
3. The manuscript is well-organized, with clear figures (especially Fig. 1–4) illustrating the motivation and the training-test gap.

**Weaknesses:**

1. The models are only fine-tuned on small datasets (6k short clips), raising concerns about robustness and scalability to large-scale or diverse domains.
2. The full pipeline (error banking, resampling, bidirectional integration) may appear complex relative to the simplicity of the LoRA tuning objective.
3. Compared with Self-Forcing[1], SVI operates on the data distribution rather than the objective level. It injects historical errors to simulate AR noise, whereas Self-Forcing constrains self-generated rollouts explicitly. This makes SVI more lightweight but potentially less principled in capturing temporal feedback dynamics.
4. As discussed in 3, while SVI effectively stabilizes autoregressive generation, its bidirectional error curation only approximates local temporal errors. It remains unclear whether the model genuinely learns long-range dependencies or simply achieves local consistency via error smoothing.

[1] Self Forcing: Bridging the Train-Test Gap in Autoregressive Video Diffusion

**Questions:**

For creative generation, how does SVI balance between temporal coherence and semantic diversity—are there cases where error correction overly smooths scene transitions?

---

> ### Author Response · Authors · 2025-11-19
> **Author Response to Weakness 1 and Weakness 2**
>
> Thank you for your insightful questions and for your thoughtful ideas about our method. We hope that our point-by-point responses address your concerns.
>
> ## [W1] Clarifying the generalization and scaling-up ability
>
> SVI can generalize well with limited training data, and also shows scaling-up ability with larger-scale data, which can be clarified in the following aspects.
>
> 1. **Generalization with limited data.** Although we only fine-tune on 6k short clips, we observe strong generalization in practice. Our evaluation data are randomly collected from the Internet rather than from tailored high-quality datasets, and they are highly diverse and exhibit a large distribution gap relative to the training set (e.g., aspect ratios, semantic content, visual quality). *On all these benchmarks, our SVI, fine-tuned on 6k clips, outperforms other long-video counterparts trained on significantly larger datasets, suggesting strong robustness and generalization.* In addition, although our training set does not contain cartoon-style data, our method generalizes well to cartoon content, including "Tom and Jerry" and "Dragon Ball" examples, as shown in the demos on our anonymous project page, further supporting its generalization ability.
>
> 2. **Scalability with more data.** In the table below, we conducted additional experiments with scaled-up training data, showing that using more data consistently improves performance across metrics. This indicates that our method not only works with small fine-tuning sets but also benefits from larger-scale training for future scaling and broader domains.
>
> | Fine-Tuning  Data | Subject  Consistency | Background  Consistency | Aesthetic  Quality | Imaging  Quality | Dynamic  Degree | Motion  Smoothness |
> |------------------|------------------------|---------------------------|----------------------|--------------------|-------------------|----------------------|
> | 0 (baseline)          | 81.44%                 | 89.81%                    | 51.33%               | 53.09%             | 61.97%            | 98.57%                  |
> | ~6,000            | 84.25%                 | 90.85%                    | 55.25%               | 59.97%             | 62.68%            | **98.69%**                  |
> | ~80,000           | **88.65%**                 | **92.38%**                    | **56.27%**               | **64.20%**             | **63.87%**            | 97.81%  |
>
> ## [W2] The full pipeline may be complex for the LoRA tuning objective
>
> We fully understand your concern, and we'd like to clarify it from two perspectives.
>
> 1. The SVI optimization objective is not overly complex for LoRA. Essentially, SVI learns image restoration with LoRA that corrects the distortions caused by DiTs. Recently, several works have explored LoRA tuning in diffusion-based image restoration [1, 2, 3], having made great progress. These works have demonstrated sufficient LoRA capacity to address complex, challenging real-world degradations, sharing a similar optimization objective with SVI: removing errors from error-corrupted inputs.
>
> 2. To further study the required capacity of LoRA, we conducted ablation studies on the LoRA rank, where the larger rank indicates more trainable parameters with larger capacity. Compared with baseline, even a small-parameter version with rank 64 already brings a clear performance gain. Moreover, when we increase the rank from our 64 to 128, the improvement is marginal, suggesting that our current LoRA has sufficient capacity to remove DiT-generated self-errors. We have updated this experiment in the revised manuscript (Table 9; Page 18).
>
> | LoRA  Rank | Subject  Consistency | Background  Consistency | Aesthetic  Quality | Imaging  Quality | Dynamic  Degree | Motion  Smoothness |
> |--------------|------------------------|---------------------------|----------------------|--------------------|-------------------|----------------------|
> | 0 (baseline) | 66.23%                 | 81.96%                    | 45.55%               | 44.29%             | **62.50%**            | 98.12%           |
> | 64           | 95.37%                 | 95.76%                    | 62.17%               | 70.34%             | 31.25%            | 98.95%               |
> | 128          | **96.10%**                 | **96.75%**                    | **62.84%**               | **70.62%**             | 25.00%            | **99.12%**               |

---

> ### Author Response · Authors · 2025-11-19
> **Author Response to Weakness 3, Weakness 4, and Question 1**
>
> ## [W3] SVI is more lightweight but potentially less principled in capturing temporal feedback dynamics than Self-Forcing
>
> We really appreciate the recognition of our method's lightweight advantage, and we would like to clarify this from two perspectives.
>
> 1. **Theoretical perspective.** During training, the effective optimized temporal horizon of SVI is no shorter than that of Self-Forcing, and therefore does not weaken the modeling of temporal feedback dynamics theoretically. Under the same 81-frame training setup, the temporal rollouts of Self-Forcing are at most 81 frames. In contrast, SVI also constrains generation to the full 81-frame sequence by injecting errors to optimize the entire video clip. From this standpoint, the two methods share a comparable temporal horizon in their training objectives for capturing temporal feedback.
>
> 2. **Experimental perspective.** To further assess temporal feedback, we compared SVI and Self-Forcing-Plus on ultra-long video generation with a long temporal horizon. Since the official Self-Forcing only supports a 1.3B backbone while ours is based on a 14B model, we adopt the latest, stronger Self-Forcing-Plus (14B) from the LightX2V Github repository [4] for a fairer, more challenging comparison. Note that Self-Forcing-Plus is a concurrent work with ours. SVI consistently outperforms Self-Forcing-Plus, empirically demonstrating its superior ability to capture temporal feedback.
>
> | Models           | Subject  Consistency | Background  Consistency | Aesthetic  Quality | Imaging  Quality | Dynamic  Degree | Motion  Smoothness |
> |------------------|------------------------|---------------------------|----------------------|--------------------|-------------------|----------------------|
> | Self-Forcing-Plus| 75.60%                 | 84.68%                    | 52.59%               | 64.62%             | **42.86%**            | 98.09%               |
> | SVI (Ours)  | **97.42%**                 | **98.03%**                    | **65.32%**               | **67.67%**            | **42.86%**            | **98.63%**               |
>
>
>
> ## [W4] Clarifying the long-range dependencies
>
>
> Thank you for your in-depth thinking and concerns. SVI can maintain long-range dependencies by optimizing the error-corrected video extension rather than merely smoothing local errors.
>
> To evaluate this long-range dependency, we assess the consistency score between different temporal horizons and the first frame in long-video generation and compare it with the latest concurrent method, Self-Forcing-Plus. For example, "1000" can measure the 1000-frame long-range dependency. The results show that SVI maintains stable performance on long-range consistency, thereby capturing long-term dependency well. We believe a key reason is that, by *explicitly encouraging the model to extend video length stably, it naturally learns to preserve coherence over long temporal spans.* The effectiveness of this type of long-video generation capability has also been demonstrated in a concurrent work [5] of us.
>
>
> | Metrics         | Method              | 1000   | 1500   | 2000   | 2500   | 3000   | 3500   | 4000   |
> |-----------------|---------------------|--------|--------|--------|--------|--------|--------|--------|
> | Long-Rang Subject Consistency      | Self-Forcing Plus  | 92.07% | 90.21% | 90.04% | 89.35% | 85.16% | 85.20% | 85.19% |
> | Long-Rang Subject Consistency      | SVI (Ours)         | **93.84%** | **93.86%** | **93.81%** | **93.82%** | **93.82%** | **93.83%** | **93.86%** |
> | Long-Rang Background Consistency   | Self-Forcing Plus  | 94.10% | 94.08% | 93.72% | 93.76% | 88.22% | 88.25% | 88.25% |
> | Long-Rang Background Consistency   | SVI (Ours)         | **95.95%** | **95.96%** | **95.29%** | **95.32%** | **95.29%** | **95.28%** | **95.31%** |
>
> ## [Q1] Balance between temporal coherence and semantic diversity
>
> We maintain temporal coherence and semantic diversity through the following strategies.
>
> 1. For temporal coherence, we replace traditional single-image image-to-video generation with motion-frame conditioning across clips (similar to the context window). By propagating motion information across neighboring clips, we can avoid “ping-pong” artifacts across clip boundaries, such as a car repeatedly moving forward and backward. This significantly improves cross-clip temporal coherence.
>
> 2. For semantic diversity, unlike existing methods that typically support only a single prompt for the entire video, SVI allows each clip to be conditioned on its own prompt. This design greatly enhances the creativity and diversity of semantics and visual content across long videos.
>
> In addition, the error correction is trained with independent video clips similar to Self-Forcing. Therefore, in theory, it shouldn't affect scene transitions noticeably. In practice, we have also not encountered this kind of failure case with an excessively smooth transition.

---

> > ### Author Response · Authors · 2025-11-19
> > **Reference List for Author Response**
> >
> > ## Reference List
> >
> > [1] Ai, Y., Huang, H., & He, R. (2024). Lora-ir: taming low-rank experts for efficient all-in-one image restoration. arXiv preprint arXiv:2410.15385.
> > [2] Park, D., Kim, H., & Chun, S. Y. (2024, September). Contribution-based low-rank adaptation with pre-training model for real image restoration. In European Conference on Computer Vision (pp. 87-105). Cham: Springer Nature Switzerland.
> > [3] Zhao, H. (2024). Efficient Image Restoration through Low-Rank Adaptation and Stable Diffusion XL. arXiv preprint arXiv:2408.17060.
> > [4] LightX2V: Light Video Generation Inference Framework
> > [5] Team, M. L., Cai, X., Huang, Q., Kang, Z., Li, H., Liang, S., ... & Zhang, T. (2025). LongCat-Video Technical Report. arXiv preprint arXiv:2510.22200.

---

> ### Comment · Reviewer_ci1H · 2025-11-24
>
> Thanks for the detailed response, which addresses many of my concerns. I still have concerns about the error simulation for long video generation, as it may fail to capture the true temporal dynamics of long videos, but it is still valuable as an additional perspective on this problem. I will consider raising my score after the rebuttal.

---

> ### Author Response · Authors · 2025-11-24
> **Follow-up Clarification about Capturing the True Temporal Dynamics of Long Videos**
>
> Dear Reviewer ci1H,
>
> We sincerely appreciate your in-depth and rigorous consideration of our method, as well as your willingness to consider raising the score. We acknowledge that, since it is trained on short video clips, the proposed error recycling cannot perfectly reconstruct the whole temporal process of long video generation, due to the limited temporal *receptive field* of the training samples (81-frame clips).
>
> However, we would like to clarify that, *compared with existing long-video solutions, our error recycling does not weaken long-range temporal modeling and, in fact, exhibits unique strengths both in principle and in practice*. Our detailed clarifications are organized as follows:
>
> 1. **Essentially, error simulation does not weaken long-range temporal modeling.** Existing long-video methods, such as Self-Forcing and FramePack, are also uniformly trained on short video clips. From this perspective, SVI shares the same limitation in short-clip training: it can only *approximate* long-video behaviors within short clips, rather than weakening the long-range dynamics of previous work.
>
> 2. **In principle, error simulation has unique strengths for approximating long-range errors.**   Compared with methods like Self-Forcing, which constrain rollouts *within an 81-frame video clip*, SVI employs a closed-loop error replay to approximate accumulated cross-clip errors over *horizons longer than 81 frames*. For example, SVI can model second-round accumulated errors when processing inputs that are already corrupted by first-round error injection. Thanks to this mechanism, SVI can simulate *longer-range behaviors and errors* under short-clip training while enjoying a *temporally longer receptive field* for cross-clip error propagation.
>
> 3. **In practice, error simulation shows clear advantages in long-range temporal consistency and dynamics.** In our experiments, SVI achieves *significantly better long-range temporal consistency and image quality* than other methods. Hence, although all methods are trained on short clips, our strategy not only offers a new perspective on error accumulation but also more faithfully approximates the long-range temporal errors in practice. In addition, qualitative comparisons further support the superior long-range consistency and temporal dynamics of SVI (please refer to the anonymous project page).
>
> | Models            | Subject Consistency | Background Consistency | Aesthetic Quality | Imaging Quality | Dynamic Degree | Motion Smoothness |
> |-------------------|---------------------|------------------------|-------------------|-----------------|----------------|-------------------|
> | FramePack         | 81.27%              | 88.29%                 | 57.52%            | 52.70%          | 0.00%          | **99.64%**        |
> | Self-Forcing-Plus | 75.60%              | 84.68%                 | 52.59%            | 64.62%          | **42.86%**     | 98.09%            |
> | SVI (Ours)        | **97.42%**          | **98.03%**             | **65.32%**        | **67.67%**      | **42.86%**     | 98.63%            |
>
> We hope that our theoretical analysis and empirical evidence can address your remaining concerns. Please let us know if you have any further questions. Thank you.
>
> Best,
> Authors of Submission #2020

---

### Official Review · Reviewer_WpZa · 2025-11-01

**Soundness:** 4
**Presentation:** 4
**Contribution:** 4
**Rating:** 8
**Confidence:** 4

**Summary:**

This paper introduces Error-Recycling Fine-Tuning (ERFT), a method designed to reduce the train–test hypothesis gap in autoregressive video diffusion models. During training, clean historical frames are typically assumed, whereas test-time generation depends on error-accumulating predictions. ERFT re-injects past model errors back into the conditioning frames, allowing the model to learn corrective behaviors. Experiments across diverse conditions and video models demonstrate strong gains in temporal stability and long-horizon video coherence.

**Strengths:**

- The paper explains the train–test mismatch and derives ERFT step-by-step in a smooth narrative. Table 4 strongly supports the reasoning behind the method.
- Figure 2 and Figure 4 present the mathematical formulation as intuitive diagrams, making the approach easy to understand even without flow matching background.
- The method improves consistency in diverse scenarios (text-to-video, audio-conditioned video, skeleton-conditioned video) and demonstrates broad applicability.
- Figure 5 and the qualitative samples show a large gap between baseline and finetuned models, especially as video length increases. The improvements in scene stability and reduction of drift are visually obvious and compelling.
- The appendix provides thoughtful analysis of current boundaries (e.g., identity consistency, scene transitions) and potential directions, giving the work strong research depth.

**Weaknesses:**

- As acknowledged by the authors (appendix B.3), the model can exhibit identity drift or swapping when the main character exits and re-enters the scene.
- The paper claims “infinite” or arbitrarily long sequences, but experiments demonstrate up to ~250 seconds. It remains uncertain how the model performs over 10+ minutes, where identity consistency, looping, or mode collapse may appear.

**Questions:**

- Appendix B.3 discusses identity limitations during scene transitions. Do you intend to develop explicit mechanisms to maintain subject consistency across transitions, and will this direction be included in the current work or future extensions?
- How stable is ERFT for extremely long generations (e.g., 10–30 minutes)?

---

> ### Author Response · Authors · 2025-11-19
>
> We really appreciate your insightful review and the encouraging assessment of our work. We hope that our point-to-point responses can address your concerns.
>
> ## [W1/Q1] ID change and our intention to solve
>
> Although SVI can mitigate low-level drift, we acknowledge that higher-level semantic identity changes remain challenging. Actually, this kind of ID swap remains an open problem in the general long-video generation community. This is the primary issue we are currently tackling.
>
> Despite these challenges, we are excited to share that SVI has the potential to relieve this kind of ID swapping by preserving general personal information. We have updated the qualitative results in the revised manuscript (Fig. 9; Page 10) and on our original anonymous project page (the link is at the end of the abstract). In this video, the woman protagonist completely exits the current scene at 30s, i.e., the end of the 6th video clip, and then re-enters for the next clip. We can find that the haircut, necklace, clothes, and general information can be kept similar to some extent, demonstrating the potential of SVI. We achieve this by fine-tuning SVI-Shot on longer video datasets to enhance long-range dependency, where the first frame in SVI-Shot inherently serves as context memory to maintain subject consistency. Extending SVI to use multiple context frames for explicit memory, thereby fully addressing identity change, is our next step.
>
> ## [W2/Q2] Extremely long generation with more than 10 minutes
>
> Thank you for your feedback. We have further evaluated end-to-end generation for over 14 minutes (more than 20,000 frames) without any human-in-the-loop post-processing. The qualitative results have been updated in the revised manuscript (Fig. 8; Page 10) and on our original anonymous project page (the link is at the end of the abstract). We leverage two practical settings to justify the effectiveness of generating infinite-length, non-looping videos. Due to the large file size of these extremely long videos, please wait a second for the webpage to load. If it still fails to load, please check our updated supplementary `.zip` file, where we have provided a backup of a highly compressed version due to size constraints. Thank you for your understanding.
>
> 1. **Human-centric generation.** We use a single long audio track (about **14 minutes**) to drive talking video generation. Each clip is conditioned on a different segment of the audio, so the motion does not loop. This example shows that SVI can maintain a stable identity for extended periods within a specific scene.
>
> 2. **Generic-domain generation.** We use an LLM to generate 200 distinct prompts and then perform end-to-end generation (**19-minute** video), conditioning each clip on its own prompt. This example demonstrates SVI’s potential to produce arbitrarily long videos across diverse, open-domain scenarios. A detailed quantitative comparison is shown below. SVI achieves the best consistency, quality, and dynamics scores. It is worth noting that in this sample, various underwater creatures (the subjects) frequently change according to the prompts, leading to an overall lower subject consistency.
>
> | Models         | Subject  Consistency | Background  Consistency | Aesthetic  Quality | Imaging  Quality | Dynamic  Degree | Motion  Smoothness |
> |---------------|---------------------|---------------------------|----------------------|--------------------|-------------------|----------------------|
> | HistoryGuidance | 61.75%            | 84.19%                    | 27.51%               | 49.65%             | 0.00%             | 99.50%               |
> | Framepacks    | 61.94%              | 78.02%                    | 44.20%               | 56.31%             | 0.00%             | **99.51%**               |
> | SVI (Ours)    | **84.75%**              | **93.80%**                    | **56.78%**               | **65.82%**             | **100.00%**           | 98.71%               |

---

### Author Response · Authors · 2025-12-01
**Summary of the Author–Reviewer Discussion**

Dear Area Chairs,

We are encouraged that the reviewers recognized our work (SVI) for its "strong performance", "broad applicability", "research depth", "interesting idea", and "clear, well‑motivated" presentation. To address the concerns raised in the reviews, we summarize the key revisions and clarifications:

1. **Stability justification:** Added extremely ultra‑long generation results (10+ minutes) and exit–reentry identity‑preservation results in Fig. 8 and Fig. 9.

2. **Extended ablations:** Added ablation studies on scale‑up ability, LoRA capacity, online error injection, and evaluations on existing benchmarks (Appendix C).

3. **Discussion of recent works:** Added discussion of concurrent methods that address the train–test gap in long‑video generation (Section 6).

4. **Clarification on capturing long‑range temporal feedback and dependencies:**
   - **Theoretical:** SVI explicitly measures accumulated temporal errors across multiple clips, providing a principled advantage for modeling temporal feedback.
   - **Empirical:** SVI outperforms other approaches on metrics targeting long‑range temporal feedback.

After the rebuttal, we received three updated reviewer feedbacks before 25 November:

- Reviewer ci1H explicitly indicated willingness to raise the score.
- Reviewer 8N9r increased the score from 6 to 8.

- Reviewer rurm maintained a positive score of 8.

Best,
Authors of Submission #2020

---

### Meta-Review · Area_Chair_YqsW · 2025-12-23

**Summary:**

This paper proposes Stable Video Infinity framework for infinite-length video generation using a DiT. In the first round, this paper received four reviews (8 8 6 4). After rebuttal, both reviewers with scores of 6 and 4 are inclined to raise their scores. Therefore, the paper is recommended for acceptance.

**Reviewer Concerns:**

After the rebuttal, the author included the reviewer's suggestions regarding long-range dependencies and comparisons on existing benchmarks.

**Reviewer Scores:**

After rebuttal, the reviewers with scores of 6 and 4 are inclined to raise their scores.

---

### Decision · Program_Chairs · 2026-01-26

Accept (Oral)